# Exploring the phenotypic consequences of tissue specific gene expression variation inferred from GWAS summary statistics

Alvaro N. Barbeira [1], Scott P. Dickinson[1], Rodrigo Bonazzola[1], Jiamao Zheng[1], Heather E. Wheeler [2,3], Jason M. Torres[4], Eric S. Torstenson[5], Kaanan P. Shah[1], Tzintzuni Garcia[6], Todd L. Edwards [7], Eli A. Stahl[8,9], Laura M. Huckins[8,9], GTEx Consortium, Dan L. Nicolae[1], Nancy J. Cox[5] & Hae Kyung Im [1]

Scalable, integrative methods to understand mechanisms that link genetic variants with phenotypes are needed. Here we derive a mathematical expression to compute PrediXcan (a gene mapping approach) results using summary data (S-PrediXcan) and show its accuracy and general robustness to misspecified reference sets. We apply this framework to 44 GTEx tissues and 100+ phenotypes from GWAS and meta-analysis studies, creating a growing public catalog of associations that seeks to capture the effects of gene expression variation on human phenotypes. Replication in an independent cohort is shown. Most of the associations are tissue specific, suggesting context specificity of the trait etiology. Colocalized significant associations in unexpected tissues underscore the need for an agnostic scanning of multiple contexts to improve our ability to detect causal regulatory mechanisms. Monogenic disease genes are enriched among significant associations for related traits, suggesting that smaller alterations of these genes may cause a spectrum of milder phenotypes.

[1] Section of Genetic Medicine, The University of Chicago, Chicago, IL 60637, USA. [2] Department of Biology, Loyola University Chicago, Chicago, IL 60660, USA. [3] Department of Computer Science, Loyola University Chicago, Chicago, IL 60660, USA. [4] Committee on Molecular Metabolism and Nutrition, The University of Chicago, Chicago, IL 60637, USA. [5] Vanderbilt Genetic Institute, Vanderbilt University Medical Center, Nashville, TN 37232, USA. [6] Center for Research Informatics, The University of Chicago, Chicago, IL 60615, USA. [7] Division of Epidemiology, Department of Medicine, Vanderbilt Genetics Institute, Vanderbilt University Medical Center, Nashville, TN 37232, USA. [8] Division of Psychiatric Genomics, Icahn School of Medicine at Mount Sinai, NYC, NY 10029, USA. [9] Department of Genetics and Genomics, Icahn School of Medicine at Mount Sinai, NYC, NY 10029, USA. A full list of consortium members appears at the end of the paper. Correspondence and requests for materials should be addressed to H.K.I. (email: haky@uchicago.edu)

Over the last decade, GWAS have been successful in robustly associating genetic loci to human complex traits. However, the mechanistic understanding of these discoveries is still limited, hampering the translation of the associations into actionable targets. Studies of enrichment of expression quantitative trait loci (eQTLs) among trait-associated variants[1–3] show the importance of gene expression regulation. Functional class quantification showed that 80% of the common variant contribution to phenotype variability in 12 diseases can be attributed to DNAase I hypersensitivity sites, further highlighting the importance of transcript regulation in determining phenotypes[4].

Many transcriptome studies have been conducted where genotypes and expression levels are assayed for a large number of individuals[5–8]. The most comprehensive transcriptome dataset, in terms of examined tissues, is the Genotype-Tissue Expression Project (GTEx): a large-scale effort where DNA and RNA were collected from multiple tissue samples from nearly 1000 individuals and sequenced to high coverage[9,10]. This remarkable resource provides a comprehensive cross-tissue survey of the functional consequences of genetic variation at the transcript level.

To integrate knowledge generated from these large-scale transcriptome studies and shed light on disease biology, we developed PrediXcan[11], a gene-level association approach that tests the mediating effects of gene expression levels on phenotypes. PrediXcan is implemented on GWAS or sequencing studies (i.e., studies with genome-wide interrogation of DNA variation and phenotypes). It imputes transcriptome levels with models trained in measured transcriptome datasets (e.g., GTEx). These predicted expression levels are then correlated with the phenotype in a gene association test that addresses some of the key limitations of GWAS[11].

Meta-analysis efforts that aggregate results from multiple GWAS have been able to identify an increasing number of associations that were not detected with smaller sample sizes[12–14]. We will refer to these results as Genome-wide association meta-analysis (GWAMA) results. In order to harness the power of these increased sample sizes while keeping the computational burden manageable, methods that use summary level data rather than individual level data are needed.

Methods similar to PrediXcan that estimate the association between intermediate gene expression levels and phenotypes, but use summary statistics have been reported: TWAS (summary version)[15] and Summary Mendelian Randomization (SMR)[16]. Another class of methods that integrate eQTL information with GWAS results are based on colocalization of eQTL and GWAS signals. Colocalized signals provide evidence of possible causal relationship between the target gene of an eQTL and the complex trait. These include RTC[1], Sherlock[17], COLOC[18], and more recently eCAVIAR[19] and ENLOC[20].

Here we derive a mathematical expression that allows us to compute the results of PrediXcan without the need to use individual-level data, greatly expanding its applicability. We compare with existing methods and outline a best practices framework to perform integrative gene mapping studies, which we term MetaXcan.

We apply the MetaXcan framework by first training over one million elastic net prediction models of gene expression traits, covering protein coding genes across 44 human tissues from GTEx, and then performing gene-level association tests over 100 phenotypes from 40 large meta-analysis consortia and dbGaP.

## Results

**Computing PrediXcan results using summary statistics**. We have derived an analytic expression to compute the outcome of PrediXcan using only summary statistics from genetic association studies. Details of the derivation are shown in the Methods section. In Fig. 1a we illustrate the mechanics of Summary-PrediXcan (S-PrediXcan) in relation to traditional GWAS and the individual-level PrediXcan method[11].

We find high concordance between PrediXcan and S-PrediXcan results indicating that in most cases, we can use the summary version without loss of power to detect associations. Figure 2 shows the comparison of PrediXcan and S-PrediXcan Z-scores for a simulated phenotype (under the null hypothesis), a cellular growth phenotype and two disease phenotypes: type 1 diabetes and bipolar disorder from the WTCCC Consortium[21]; see Supplementary Notes 1, 2 and 3 for details. For the simulated phenotype, the study sets (in which GWAS is performed) and the reference set (in which LD between SNPs is computed) were African, East Asian, and European subsets from 1000 Genomes. The training set (in which prediction models are trained) was European (DGN Cohort[5]) in all cases. The high correlation between PrediXcan and S-PrediXcan demonstrates the robustness of our method to mismatches between reference and study sets. Despite the generally good concordance between the summary and individual level methods, there were a handful of false positive results with S-PrediXcan much more significant than PrediXcan. This underscores the need to use closely matched LD information whenever possible. Supplementary Fig. 11 shows S-PrediXcan's performance on a phenotype simulated under the alternative hypothesis.

Notice that we are not testing here whether PrediXcan itself is robust to population differences between training and study sets. Robustness of the prediction across populations has been previously reported[22]. We further corroborated this in Supplementary Fig. 10.

Next we compare with other summary result-based methods such as S-TWAS, SMR, and COLOC.

**Colocalization estimates complement PrediXcan results**. One class of methods seeks to determine whether eQTL and GWAS signals are colocalized or are distinct although linked by LD. This class includes COLOC[18], Sherlock[17], and RTC[1], and more recently eCAVIAR[19], and ENLOC[20]. Thorough comparison between these methods can be found in refs. [18,19]. HEIDI, the post filtering step in SMR that estimates heterogeneity of GWAS and eQTL signals, can be included in this class. We focus here on COLOC, whose quantification of the probability of five configurations complements well with S-PrediXcan results.

COLOC provides the probability of five hypotheses: H0 corresponds to no eQTL and no GWAS association, H1 and H2 correspond to association with eQTL but no GWAS association or vice-versa, H3 corresponds to eQTL and GWAS association but independent signals, and finally H4 corresponds to shared eQTL and GWAS association. P0, P1, P2, P3, and P4 are the corresponding probabilities for each configuration. The sum of the five probabilities is 1. The authors[18] recommend to interpret H0, H1, and H2 as limited power; for convenience we will aggregate these three hypotheses into one event with probability 1-P3-P4.

Figure 3 shows ternary plots[23] with P3, P4, and 1-P3-P4 as vertices. The blue region, top subtriangle, corresponds to high probability of colocalized eQTL and GWAS signals (P4). The orange region at bottom left corresponds to high probability of distinct eQTL and GWAS signals (P3). The gray region at center and bottom right corresponds to low probability of both colocalization and independent signals.

Figure 3b shows association results for all gene-tissue pairs with the height phenotype. We find that most associations fall in the gray, "undetermined," region. When we restrict the plot to S-

PrediXcan Bonferroni-significant genes (Fig. 3c), three distinct peaks emerge in the high P4 region (P4 > 0.5, "colocalized signals"), high P3 region (P3 > 0.5, "independent signals" or "non-colocalized signals"), and "undetermined" region. Moreover, when genes with low prediction performance are excluded (Supplementary Fig. 6d) the "undetermined" peak significantly diminishes.

These clusters provide a natural way to classify significant genes and complement S-PrediXcan results. Depending on false positive/false negative trade-off choices, genes in the "independent signals" or both "independent signals" and "undetermined" can be filtered out. The proportion of colocalized associations (P4 > 0.5) ranged from 5 to 100% depending on phenotype with a median of 27.6%. The proportion of "non-colocalized"

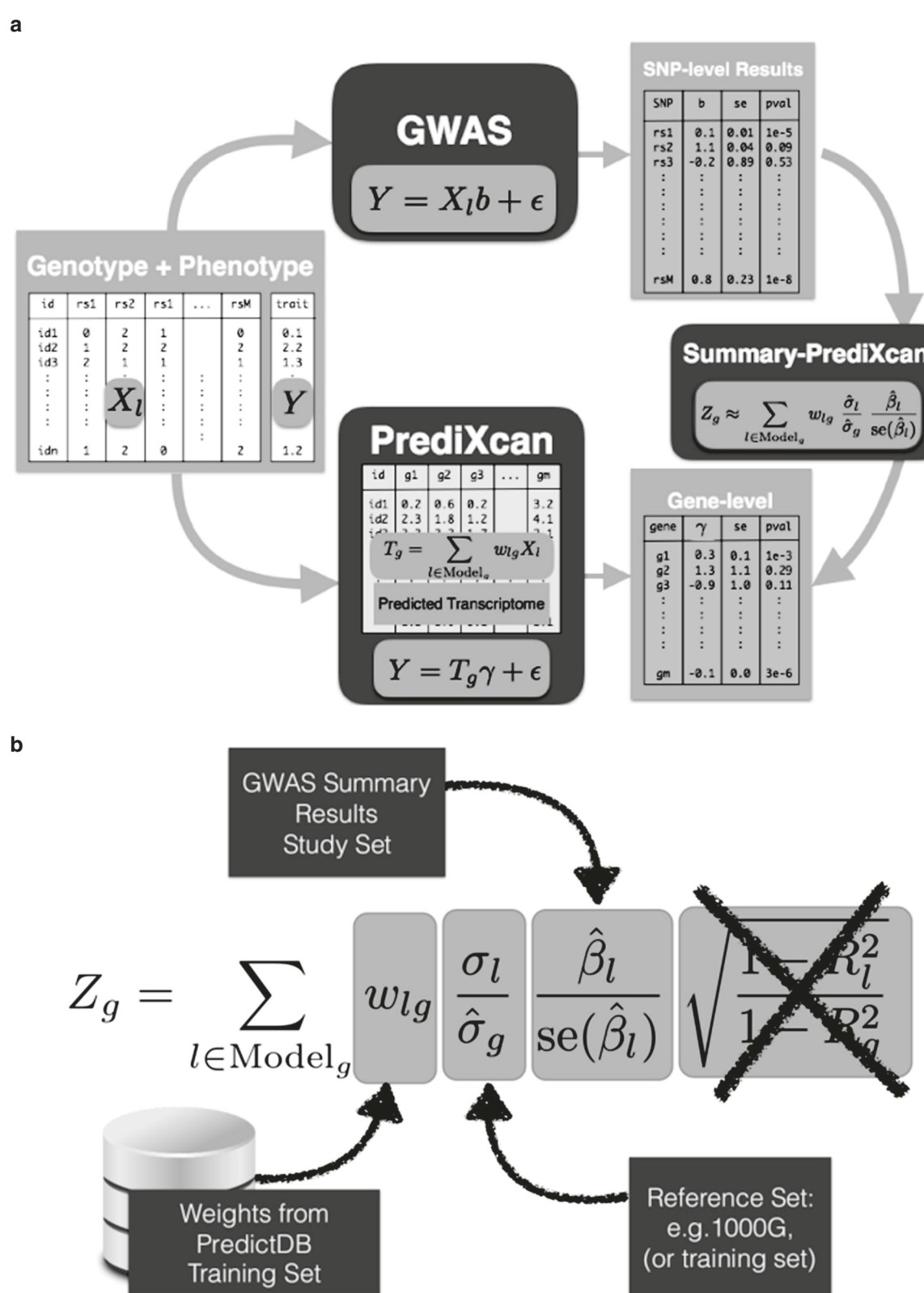

associations ranged from 0 to 77% with a median of 27.0%. Supplementary Table 1 summarizes the percentages of significant associations that fall into the different colocalization regions.

This post-filtering idea was first implemented in the SMR approach using HEIDI. Comparison of COLOC results with HEIDI is shown in Supplementary Fig. 6e to 6h.

**Comparison of S-PrediXcan to S-TWAS.** Gusev et al. have proposed Transcriptome-Wide Association Study based on summary statistics (STWAS), which imputes the SNP level $Z$-scores into gene level $Z$-scores. This is not the same as computing the results of individual level TWAS. We show (in Methods section) that the difference between the individual level and summary level TWAS is given by the factor $\sqrt{\frac{1-R_l^2}{1-R_g^2}}$, where $R_l$ is the proportion of variance in the phenotype explained by a SNP's allelic dosage, and $R_g$ is the proportion explained by gene expression (see Methods section). For most practical purposes we have found that this factor is very close to 1 so that if the same prediction models were used, no substantial difference between S-TWAS and S-PrediXcan should be expected.

Figure 4a shows a diagram of S-PrediXcan and S-TWAS. Both use SNP to phenotype associations results ($Z_{X,Y}$) and prediction weights ($w_{X,Tg}$) to infer the association between the gene expression level ($T_g$) and phenotype ($Y$).

Figure 4b compares S-TWAS significance (as reported in ref. [24]) to S-PrediXcan significance. The difference between the two approaches is mostly driven by the different prediction models: TWAS uses BSLMM[25] whereas PrediXcan uses elastic net[26]. BSLMM allows two components: one sparse (small set of large effect predictors) and one polygenic (all variants contribute some marginal effect to the prediction). For PrediXcan we have chosen to use a sparse model (elastic net) based on the finding that the genetic component of gene expression levels is mostly sparse[27].

Figure 4c shows that the COLOC-estimated proportion of non-colocalized (independent) GWAS and eQTL signals is larger among TWAS significant genes than among S-PrediXcan significant ones. We believe this is due to the polygenic component of BSLMM models, a wider set of SNPs increasing the chances of COLOC yielding a non-colocalized result. Figure 4d shows that, for most traits, the COLOC-estimated proportion of colocalized signals is larger among S-PrediXcan significant genes than S-TWAS significant genes.

**Comparison of S-PrediXcan to SMR.** Zhu et al. have proposed Summary Mendelian Randomization (SMR)[16], a summary data based Mendelian randomization that integrates eQTL results to determine target genes of complex trait-associated GWAS loci. They derive an approximate $\chi_1^2$-statistic (Eq 5 in ref. [16]) for the mediating effect of the target gene expression on the phenotype. Figure 5a depicts this mechanism.

Unfortunately, the derived statistic is not well calibrated. A QQ plot comparing the SMR statistic (under the null hypothesis of genome-wide significant eQTL signal and no GWAS association) shows deflation. The sample mean of the statistic is ≈0.93 instead of 1, the expected value for the mean of a $\chi_1^2$ random variable (see Fig. 5e, f and Methods section for details). The $\chi^2$ approximation is only valid in two extreme cases: when the eQTL association is much stronger than the GWAS association or vice versa, when the GWAS association is much stronger than the eQTL association (see Methods section for details).

One limitation is that the significance of the SMR statistic is the lower of the top eQTL association (genotype to expression) or the GWAS association (genotype to phenotype) as shown in Fig. 5e, f. Given the much larger sample sizes of GWAS studies, for most genes, the combined significance will be determined by the eQTL association. The combined statistic forces us to apply multiple testing correction for all genes, even those that are distant to GWAS associated loci, which is unnecessarily conservative. Keep in mind that currently both SMR and PrediXcan only use *cis* associations. An example may clarify this further. Let us suppose that for a given phenotype there is only one causal SNP and that the GWAS yielded a highly significant p-value, say $10^{-50}$. Let us also suppose that there is only one gene (gene A) in the vicinity (we are only using *cis* predictors) associated with the causal SNP with $p = 10^{-5}$. SMR would compute the p-values of all genes and yield a p-value ≈$10^{-5}$ for gene A (the less significant p-value). However, after multiple correction this gene would not be significantly associated with the phenotype. Here it is clear that we should not be adjusting for testing of all genes when we know a priori that only one is likely to produce a gene level association. In contrast, the PrediXcan p-value would be ≈$10^{-50}$ for gene A and would be distributed uniformly from 0 to 1 for the remaining genes. Most likely only gene A (or perhaps a handful of genes, just by chance) would be significant after Bonferroni correction. If we further correct for prediction uncertainty (here = eQTL association), a p-value of ≈$10^{-5}$ would remain significant since we only need to correct for the (at most) handful of genes that were Bonferroni significant for the PrediXcan p-value.

Another potential disadvantage of this method is that only top-eQTLs are used for testing the gene level association. This does not allow to aggregate the effect on the gene across multiple variants.

Figure 5b compares S-PrediXcan (elastic net) and SMR association results. As expected, SMR p-values tend to be less significant than S-PrediXcan's in large part due to the additional adjustment for the uncertainty in the eQTL association. Figure 5c, d show that the SMR significance is bounded by the eQTL and GWAS association strengths of the top eQTL. Figure 5g shows a comparison between SMR's and S-PrediXcan's proportion of non-colocalization, while Fig. 5h compares proportion of colocalization, as estimated by COLOC. SMR shows a higher proportion of colocalized and independent signals. This is expected since SMR uses a more stringent eQTL association

**Fig. 1** Comparison between GWAS, PrediXcan, and S-PrediXcan. **a** Compares GWAS, PrediXcan, and Summary-PrediXcan. Both GWAS and PrediXcan take genotype and phenotype data as input. GWAS computes the regression coefficients of $Y$ on $X_l$ using the model $Y = a + X_l b + \epsilon$, where $Y$ is the phenotype and $X_l$ the individual SNP dosage. The output is a table of SNP-level results. PrediXcan, in contrast, starts first by predicting/imputing the transcriptome. Then it calculates the regression coefficients of the phenotype $Y$ on each gene's predicted expression $T_g$. The output is a table of gene-level results. Summary-PrediXcan directly computes the gene-level association results using the output from GWAS. **b** Shows the components of the formula to calculate PrediXcan gene-level association results using summary statistics. The different sets involved as input data are shown. The regression coefficient between the phenotype and the genotype is obtained from the study set. The training set is the reference transcriptome dataset where the prediction models of gene expression levels are trained. The reference set (1000G, or training set having some advantages) is used to compute the variances and covariances (LD structure) of the markers used in the predicted expression levels. Both the reference set and training set values are precomputed and provided to the user so that only the study set results need to be provided to the software. The crossed out term was set to 1 as an approximation. We found this approximation to have negligible impact on the results

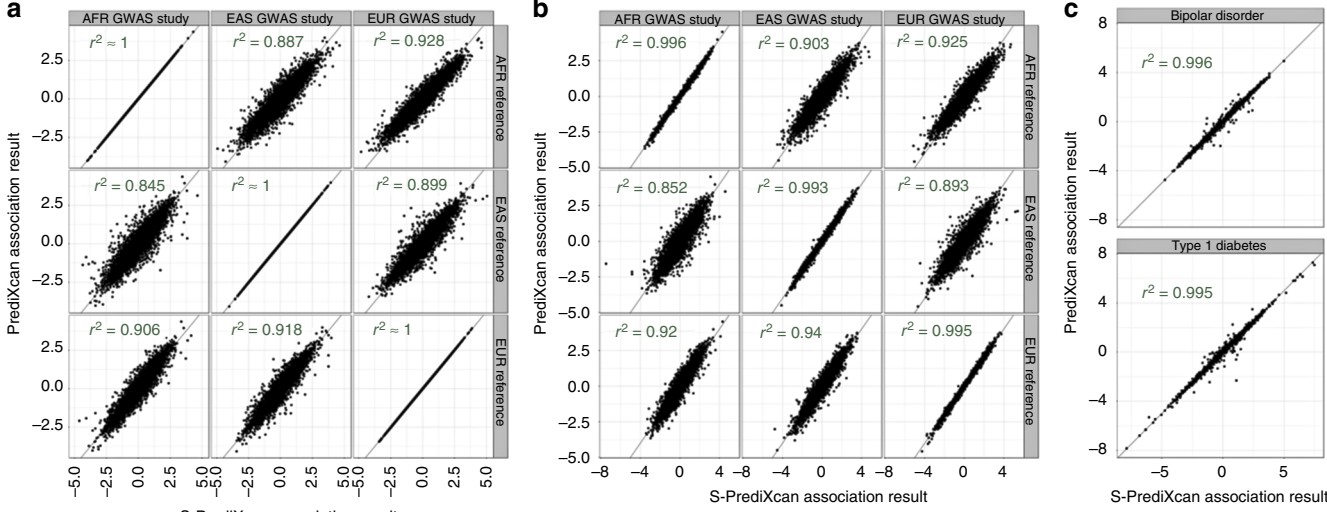

**Fig. 2** Comparison of PrediXcan and S-PrediXcan results in real and simulated traits. This figure shows a comparison of PrediXcan vs. S-PrediXcan for **a** a simulated phenotype under null hypothesis of no genetic component; **b** a cellular phenotype (=intrinsic growth); and **c** bipolar disorder and type 1 diabetes studies from Wellcome Trust Case Control Consortium (WTCCC). Gene expression prediction models were based on the DGN cohort presented in ref. [11]. For the simulated phenotype, study sets (GWAS set) and reference sets (LD calculation set) consisted of African (661), East Asian (504), and European (503) individuals from the 1000 Genomes Project. When the same study set is used as reference set, we obtained a high correlation (coefficient of determination): $r^2 > 0.99999$. For the intrinsic growth phenotype, study sets were a subset of 140 individuals from each of the African, Asian, and European groups from 1000 Genomes Project. The reference set was the same as for the simulated phenotype. For the disease phenotypes, the study set consisted of British individuals, and the LD calculation set was the European population subset of the 1000 Genomes Project

criterion so that there are few significant genes in the undetermined region.

SMR introduces a post filtering step via an approach called HEIDI, which is compared to COLOC in Fig. 3e and Supplementary Fig. 6.

**MetaXcan framework**. Building on S-PrediXcan and existing approaches, we define a general framework (MetaXcan) to integrate eQTL information with GWAS results and map disease-associated genes. This evolving framework can incorporate models and methods to increase the power to detect causal genes and filter out false positives. Existing methods fit within this general framework as instances or components (Fig. 6a).

The framework starts with the training of prediction models for gene expression traits followed by a selection of high-performing models. Next, a mathematical operation is performed to compute the association between each gene and the down-stream complex trait. Additional adjustment for the uncertainty in the prediction model can be added. To avoid capturing LD-contaminated associations, which can occur when expression predictor SNPs and phenotype causal SNPs are different but in LD, we use colocalization methods that estimate the probability of shared or independent signals.

PrediXcan implementations use elastic net models motivated by our observation that gene expression variation is mostly driven by sparse components[27]. TWAS implementations have used Bayesian Sparse Linear Mixed Models[25] (BSLMM). SMR fits into this scheme with prediction models consisting solely of the top eQTL for each gene (weights are not necessary here since only one SNP is used at a time).

For the last step, we chose COLOC to estimate the probability of colocalization of GWAS and eQTL signals. COLOC probabilities cluster more distinctly into different classes and thus, unlike other methods, suggests a natural cut off threshold at $P = 0.5$. Another advantage of COLOC is that for genes with low probability of colocalization, it further distinguishes distinct

GWAS and eQTL signals from low power. This is a useful feature that future development of colocalization methods should also offer. SMR, on the other hand, uses its own estimate of "heterogeneity" of signals calculated by HEIDI.

**Suggested association analysis pipeline**.

1. Perform PrediXcan or S-PrediXcan using all tissues. Use Bonferroni correction for all gene-tissue pairs: keep $p < 0.05/$number of gene-tissue pairs tested.
2. Keep associations with significant prediction performance adjusting for number of PrediXcan significant gene-tissue pairs: keep prediction performance $p$-values $< 0.05/$(number of significant associations from previous step).
3. Filter out LD-contaminated associations, i.e., gene-tissue pairs in the "independent signal" (="non-colocalized") region of the ternary plot (See Fig. 3a): keep COLOC P3 < 0.5 (Blue and gray regions in Fig. 3a).
4. If further reduction of number of genes to be taken to replication or validation is desired, keep only hits with explicit evidence of colocalization: P4 > 0.5 (Blue region in Fig. 3a).

Any choice of thresholds has some level of arbitrariness. Depending on the false positive and negative trade off, these numbers may be changed.

**Gene expression variation is associated to diverse traits**. We downloaded summary statistics from meta analyses of over 100 phenotypes from 40 consortia. The full list of consortia and phenotypes is shown in Supplementary Data 2. We tested association between these phenotypes and the predicted expression levels using elastic net models in 44 human tissues from GTEx as described in the Methods section, and a whole blood model from the DGN cohort presented in ref. [11]. We illustrate this application in Fig. 6b.

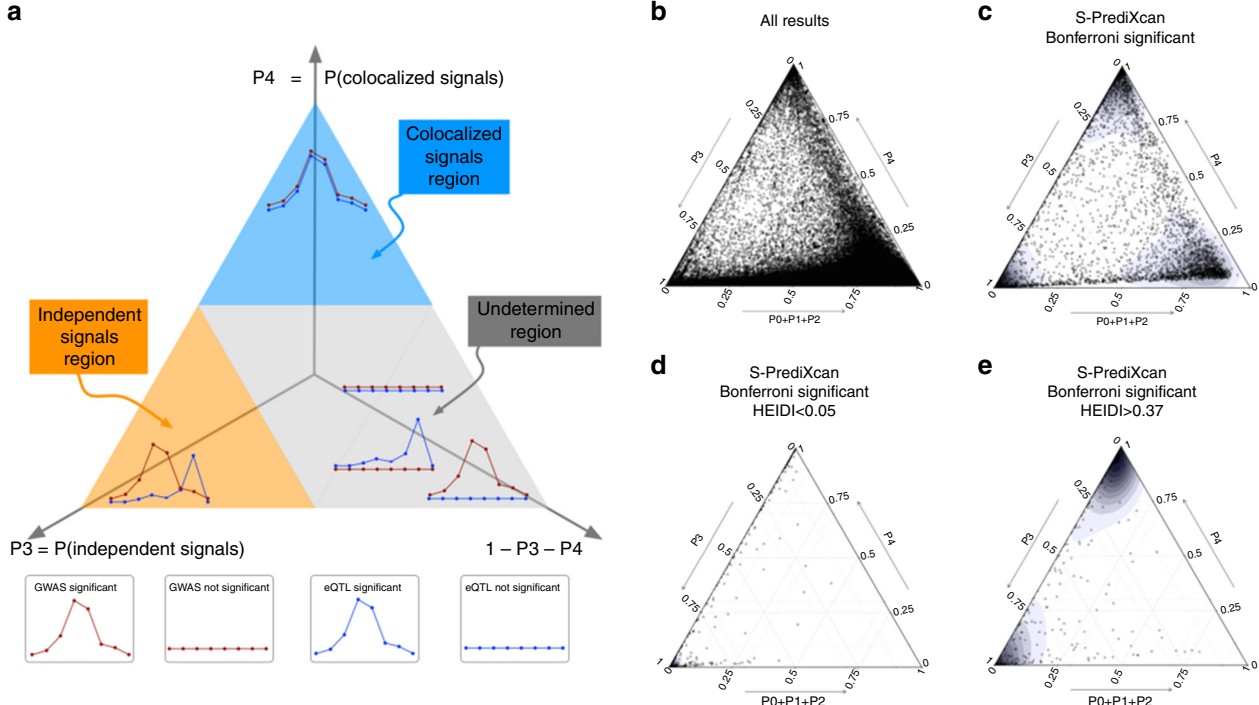

**Fig. 3** Colocalization status of S-PrediXcan results. **a** Shows a ternary plot that represents the probabilities of various configurations from COLOC. This plot conveniently constrains the values such that the sum of the probabilities is 1. All points in a horizontal line have the same probability of "colocalized" GWAS and eQTL signals (P4), points on a line parallel to the right side of the triangle (NW to SE) have the same probability of "Independent signals" (P3), and lines parallel to the left side of the triangle (NE to SW) correspond to constant P0+P1+P2. Top sub-triangle in blue corresponds to high probability of colocalization (P4 > 0.5), lower left sub-triangle in orange corresponds to probability of independent signals (P3 > 0.5), and lower right parallelogram corresponds to genes without enough power to determine or reject colocalization. The following panels present ternary plots of COLOC probabilities with a density overlay for S-PrediXcan results of the Height phenotype. **b** Shows the colocalization probabilities for all gene-tissue pairs. Most results fall into the "undetermined" region. **c** Shows that if we keep only Bonferroni-significant S-PrediXcan results, associations tend to cluster into three distinct regions: "independent signals," "colocalized," and "undetermined." **d** Shows that HEIDI significant genes (to be interpreted as high heterogeneity between GWAS and eQTL signals, i.e., distinct signals) tightly cluster in the "independent signal" region, in concordance with COLOC. A few genes fall in the "colocalized" region, in disagreement with COLOC classification. Unlike COLOC results, HEIDI does not partition the genes into distinct clusters and an arbitrary cutoff *p*-value has to be chosen. **e** Shows genes with large HEIDI *p*-value (no evidence of heterogeneity) which fall in large part in the "colocalized" region. However a substantial number fall in "independent signal" region, disagreeing with COLOC's classification

S-PrediXcan's results tend to be more significant as the genetic component of gene expression increases (larger cross-validated prediction performance $R^2$). Similarly, S-PrediXcan associations tend to be more significant when prediction is more reliable (*p*-values of association between predicted and observed expression levels are more significant, i.e., when prediction performance *p*-value is smaller). The trend is seen both when results are averaged across all tissues for a given phenotype or across all phenotypes for a given tissue, as displayed in Supplementary Figs. 1-4. This trend was also robust across different monotonic functions of the *Z*-scores.

We used a Bonferroni threshold accounting for all the gene-tissue pairs that were tested (0.05/total number of gene-tissue pairs ≈2.5e-7). This approach is conservative because the correlation between tissues would make the total number of independent tests smaller than the total number of gene-tissue pairs. Height had the largest number of significantly associated unique genes at 1686 (based on a GWAMA of 250 K individuals). Other polygenic diseases with a large number of associations include schizophrenia with 305 unique significant genes (*n* = 150 K individuals), low-density lipoprotein cholesterol (LDL-C) levels with 296 unique significant genes (*n* = 188 K), other lipid levels, glycemic traits, and immune/inflammatory disorders such as rheumatoid arthritis and inflammatory bowel disease. For other psychiatric phenotypes, a much smaller number of

significant associations was found, with eight significant genes for bipolar disorder (*n* = 16,731) and one for major depressive disorder (*n* = 18,759), probably due to smaller sample sizes, but also smaller effect sizes.

When step 2 from the suggested pipeline is applied, keeping only reliably predicted genes, we are left with 739 genes for height, 150 for schizophrenia, 117 for LDL-C levels.

After step 3, which keeps genes that are without strong evidence of LD-contamination, these numbers dropped to 264 for height, 58 for schizophrenia, and 60 for LDL-C levels. After step 4, which keeps only genes with strong evidence of colocalization, we find 215 genes for height, 49 for schizophrenia, and 35 for LDL-C. The counts for the full set of phenotypes can be found in Supplementary Data 2.

Mostly, genome-wide significant genes tend to cluster around known SNP-level genome-wide significant loci or sub-genome-wide significant loci. Regions with sub-genome-wide significant SNPs can yield genome-wide significant results in S-PrediXcan, because of the reduction in multiple testing and the increase in power arising from the combined effects of multiple variants. Supplementary Table 2 lists a few examples where this occurs.

The proportion of colocalized associations (P4 > 0.5) ranged from 5 to 100% depending on phenotype with a median of 27.6%. The proportion of "non colocalized" associations ranged from 0 to 77% with a median of 27.0%.

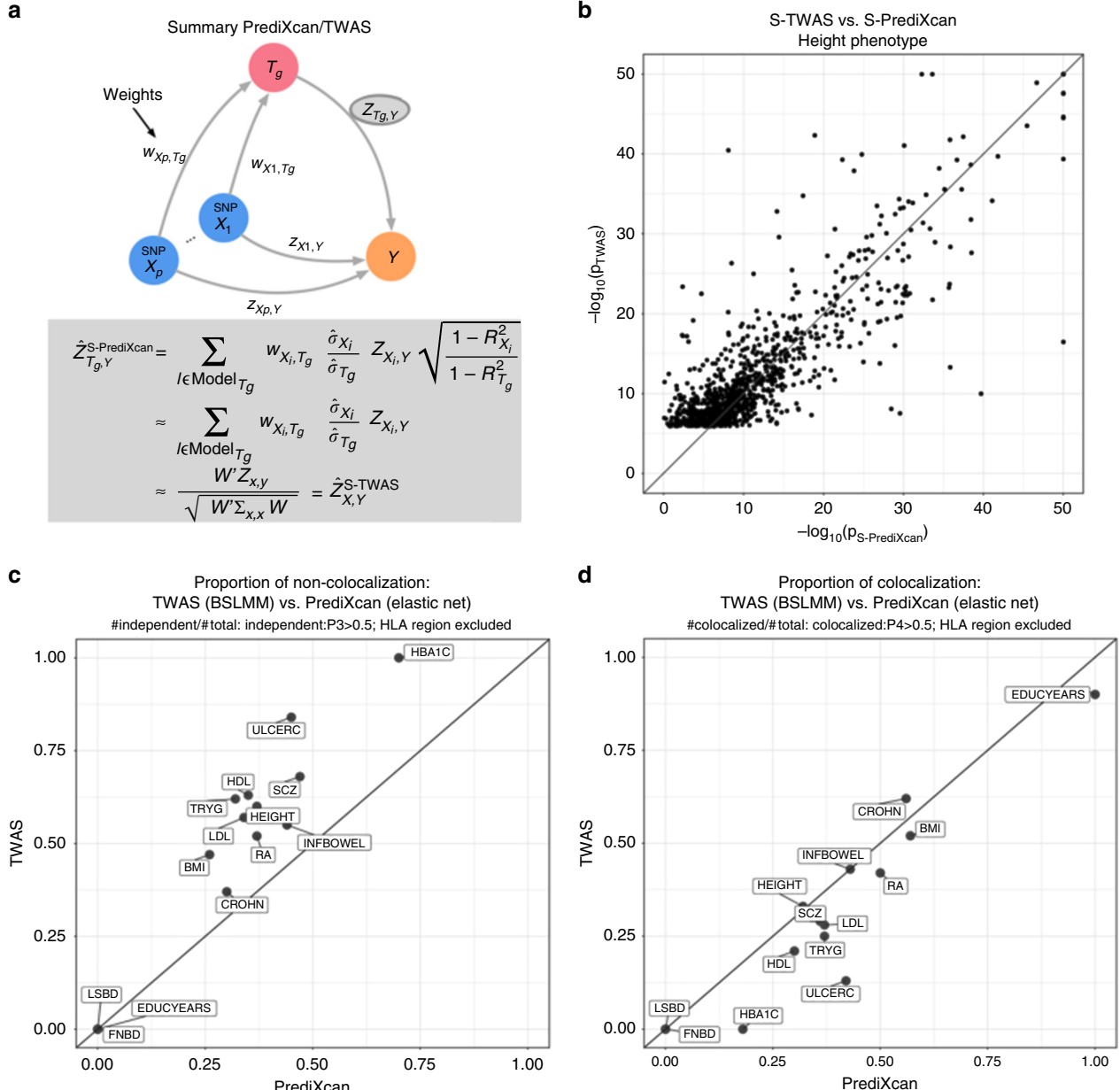

**Fig. 4** Comparison between S-PrediXcan and S-TWAS. **a** Depicts how summary-TWAS and PrediXcan test the mediating role of gene expression level $T_g$. Multiple SNPs are linked to the expression level of a gene via weights $w_{X,Tg}$. **b** Shows the significance of Summary-TWAS (BSLMM) vs. summary-PrediXcan (elastic net), for the height phenotype across 44 GTEx tissues. There is a small bias caused by using S-TWAS results available from[24], which only lists significant hits. S-PrediXcan tends to yield a larger number of significant associations (see Supplementary Fig. 12). P-values were thresholded at 10−50 for visualization purposes. **c** Shows the proportion of non-colocalized associations (distinct eQTL and GWAS signals) from S-TWAS significant vs. S-PrediXcan significant results. For all phenotypes, S-TWAS has a higher proportion of LD-contaminated signals compared to S-PrediXcan, as estimated via COLOC. **d** Shows the proportion of colocalized associations (shared eQTL and GWAS signals) from S-TWAS significant vs. S-PrediXcan significant results. For most phenotypes, TWAS has lower proportion of colocalized signals compared to S-PrediXcan, as estimated via COLOC. Phenotype abbreviations are as follows: FNBD Femoral Neck Bone Density, LSBD Lumbar Spine Bone Density, BMI Body Mass Index, HEIGHT Height, LDL Low-Density Lipoprotein Cholesterol, HDL High-Density Lipoprotein Cholesterol, TRYG Tryglicerides, CROHN Crohn's Disease, INFBOWEL Inflammatory Bowel's Disease, ULCERC Ulcerative Colitis, HBA1C Hemogoblin Levels, HOMA-IR HOMA Insulin Response, SCZ Schizophrenia, RA Rheumatoid Arthritis, COLLEGE College Completion, EDUCYEARS Education Years

See full set of results in our online catalog (gene2pheno.org). Significant gene-tissue pairs are included in Supplementary Data 3. To facilitate comparison, the catalog contains all SMR results we generated and the S-TWAS results reported by ref. [24] for 30 GWAS traits and GTEx BSLMM models. Note that SMR application to 28 phenotypes was reported by ref. [28] using whole blood eQTL results from ref. [29].

**Small gene expression changes associated to mild phenotypes.** We reasoned that if complete knock out of monogenic disease genes cause severe forms of the disease, more moderate alterations of gene expression levels (as affected by regulatory variation in the population) could cause more moderate forms of the disease. Thus moderate alterations in expression levels of monogenic disease genes (such as those driven by eQTLs) may have an effect

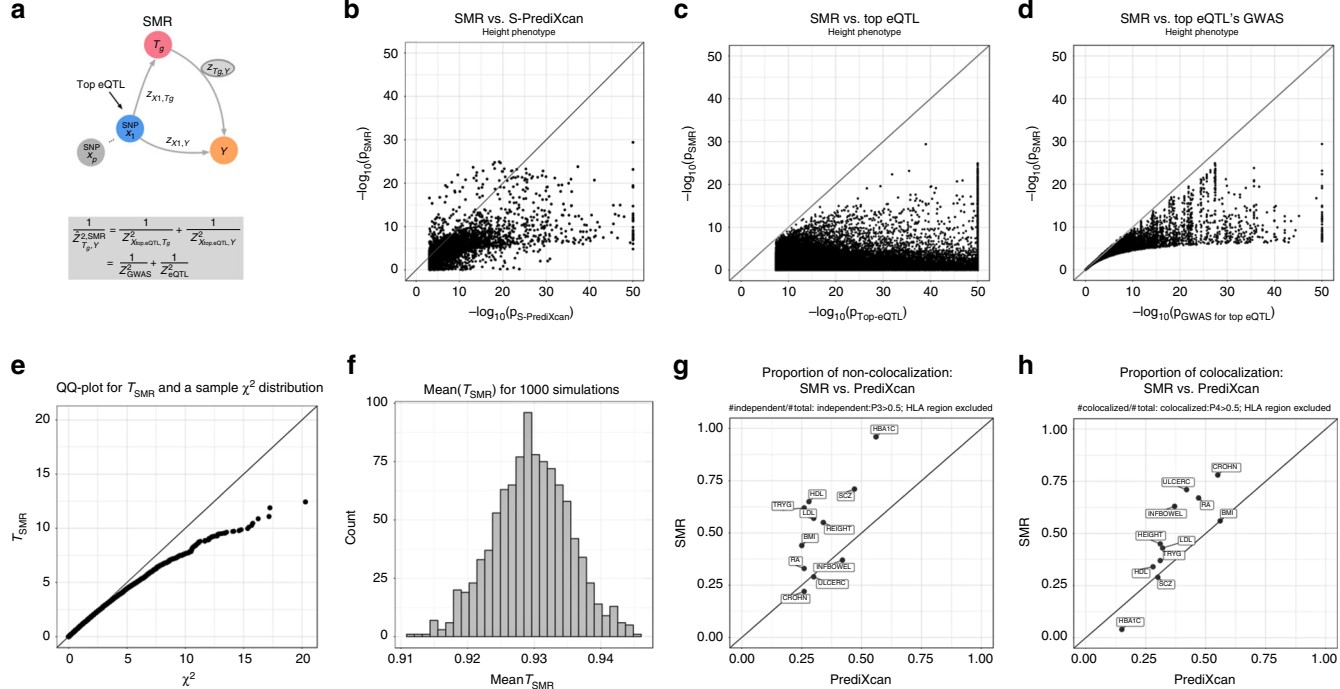

**Fig. 5** Comparison between summary-PrediXcan and SMR. **a** Depicts how SMR tests the mediating role of gene expression level $T_g$. The top eQTL is linked to the phenotype as an instrumental variable in a Mendelian Randomization approach. **b** Shows the significance of SMR vs. the significance of Summary-PrediXcan. As expected, SMR associations tend to be smaller than S-PrediXcan ones. **c** and **d** show that the SMR statistics significance is bounded by GWAS and eQTL p-values. The p-values (−log10) of the SMR statistics are plotted against the GWAS p-value of the top eQTL SNP (**c**), and the gene's top eQTL p-value (**d**). **e** Shows a QQ plot for simulated values of $T_{SMR}$. Under the null hypothesis of significant eQTL signal and no GWAS association, we generated random values for $Z_{GWAS}^2$ and $Z_{eQTL}^2$ following the simulations from ref. [16]. $T_{SMR}$ statistic was calculated from these values, and compared to a $\chi_1^2$ distribution to illustrate this statistics' deflation. **f** shows the sample mean of $T_{SMR}$ from 1000 simulations, centered close to 0.93, instead of the expected value of 1 for a $\chi_1^2$-distributed variable. **g** shows the proportion of non-colocalized significant associations to total significant associations in PrediXcan and SMR. **h** Shows the proportion of colocalized significant associations (shared eQTL and GWAS signals). As expected, SMR shows a higher proportion of colocalized and non-colocalized associations than PrediXcan. This is caused by SMR's high eQTL significance threshold, that rules out most of the genes with low colocalization power (P0 + P1 + P2 > 0.5). For some of the associations, GWAS and eQTL p-values were more significant than shown since they were thresholded at $10^{-50}$ to improve visualization. Phenotype abbreviations are as follows: FNBD Femoral Neck Bone Density, LSBD Lumbar Spine Bone Density, BMI Body Mass Index, HEIGHT Height, LDL Low-Density Lipoprotein Cholesterol, HDL High-Density Lipoprotein Cholesterol, TRYG Tryglicerides, CROHN Crohn's Disease, INFBOWEL Inflammatory Bowel's Disease, ULCERC Ulcerative Colitis, HBA1C Hemogoblin Levels, HOMA-IR HOMA Insulin Response, SCZ Schizophrenia, RA Rheumatoid Arthritis, COLLEGE College Completion, EDUCYEARS Education Years

on related complex traits, and this effect could be captured by S-PrediXcan association statistics. To test this hypothesis, we obtained genes listed in the ClinVar database[30] for obesity, rheumatoid arthritis, diabetes, Alzheimer's, Crohn's disease, ulcerative colitis, age-related macular degeneration, schizophrenia, and autism. Figure 7 displays the QQ plot for all associations and compares to those in ClinVar database. As postulated, we found enrichment of significant S-PrediXcan associations for ClinVar genes for all tested phenotypes except for autism and schizophrenia. The lack of significance for autism is probably due to insufficient power: the distribution of p-values is close to the null distribution. In contrast, for schizophrenia, many genes were found to be significant in the S-PrediXcan analysis. There are several reasons that may explain this lack of enrichment: genes identified with GWAS and subsequently with S-PrediXcan have rather small effect sizes, so that it would not be surprising that they were missed until very large sample sizes were aggregated; ClinVar genes may originate from rare mutations that are not well covered by our prediction models, which are based on common variation (due to limited sample sizes of eQTL studies and the minor allele frequency –MAF filter used in GWAS studies); or the mechanism of action of the schizophrenia linked ClinVar genes may be different than the alteration of expression levels. Also, the pathogenicity of some of the ClinVar

entries has been questioned[31]. The list of diseases in ClinVar used to generate the enrichment figures can be found in Supplementary Data 1, along with the corresponding association results.

**Agnostic scanning across GTEx tissues improves discovery.** Most genes were found to be significantly associated in a handful of tissues as illustrated in Fig. 8b. For example, for LDL-C levels, liver was the most enriched tissue in significant associations as expected given known biology of this trait (See Supplementary Fig. 5). This prominent role of liver was apparent despite the smaller sample size available for building liver models ($n = 97$), which was less than a third of the numbers available for muscle ($n = 361$) or lung ($n = 278$).

However, in general, tissues expected to stand out as more enriched for diseases given currently known biology did not consistently do so when we looked at the average across all (significant) genes, using various measures of enrichment. For example, the enrichment in liver was less apparent for high-density lipoprotein cholesterol (HDL-C) or triglyceride levels. We find for many significant associations that the evidence is present across multiple tissues. This may be caused by a combination of context specificity and sharing of regulatory mechanism across tissues.

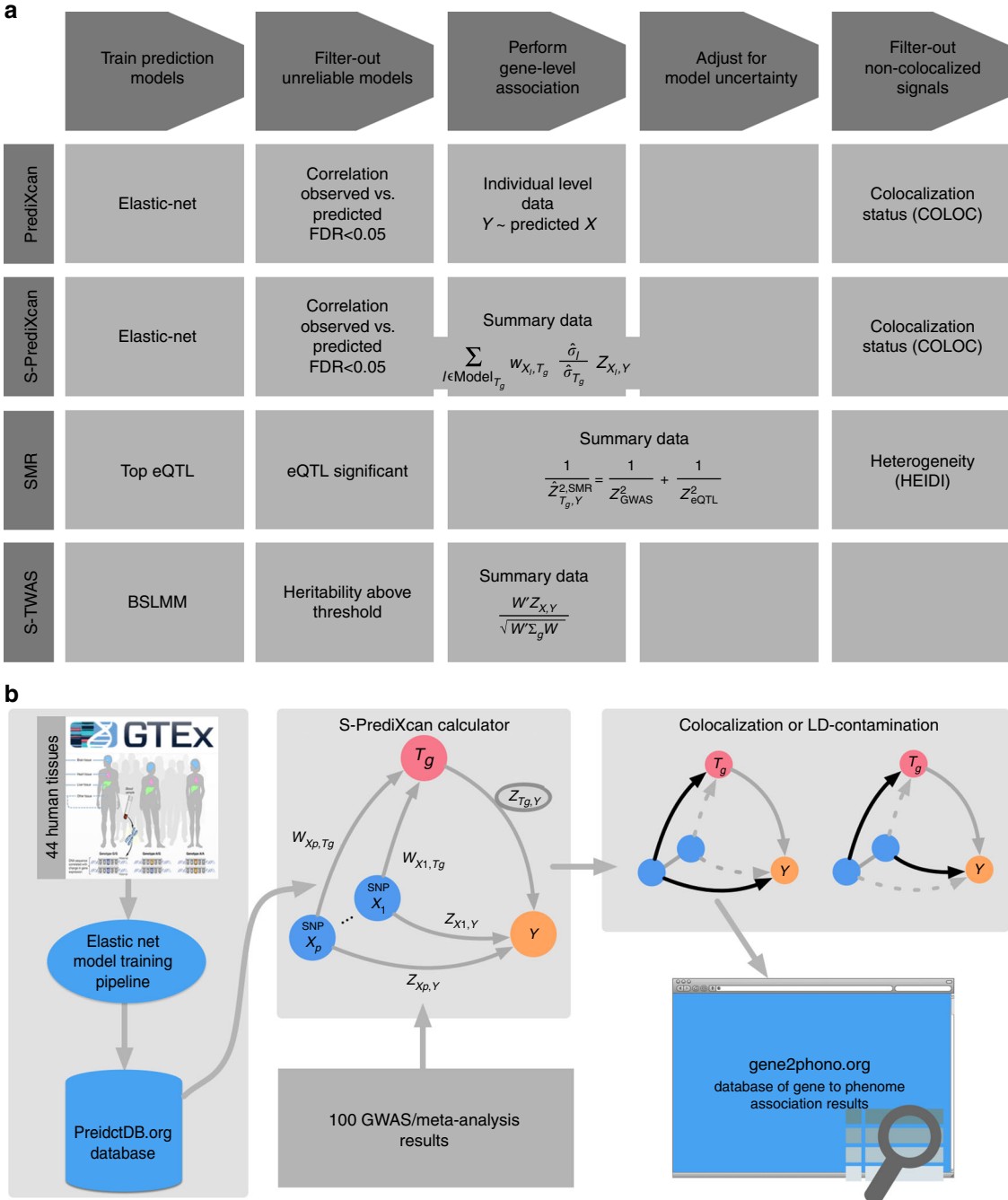

**Fig. 6** MetaXcan framework and application. **a** Shows a general framework (MetaXcan) which encompasses methods such as PrediXcan, TWAS, SMR, COLOC among others. **b** Summarizes the application of the MetaXcan framework with S-PrediXcan using 44 GTEx tissue transcriptomes and over 100 GWAS and meta analysis results. We trained prediction models using elastic-net[26] and deposited the weights and SNP covariances in the publicly available resource (http://predictdb.org/). The weights, covariances, and over 100 GWAS summary results were processed with S-PrediXcan. Colocalization status was computed and the full set of results was deposited in gene2pheno.org

Next, we illustrate the challenges of identifying disease relevant tissues based on eQTL information using three genes with well established biology: *C4A* for schizophrenia[32] and *SORT1*[33] and *PCSK9* both for LDL-C and cardiovascular disease. S-PrediXcan results for these genes and traits, and regulatory activity by tissue (as measured by the proportion of expression explained by the genetic component), are shown in Fig. 8a. Representative results are shown in Supplementary Tables 3, 4 and 5. Supplementary Data 4 contains the full set of MetaXcan results (i.e., association, colocalization, and HEIDI) for these genes.

*SORT1* is a gene with strong evidence for a causal role in LDL-C levels, and as a consequence, is likely to affect risk for cardiovascular disease[33]. This gene is most actively regulated in liver (close to 50% of the expression level of this gene is determined by the genetic component) with the most significant S-PrediXcan association in liver (*p*-value $\approx 0$, $Z = -28.8$), consistent with our prior knowledge of lipid metabolism. In this example, tissue specific results suggest a causal role of *SORT1* in liver.

However, in the following example, association results across multiple tissues do not allow us to discriminate the tissue

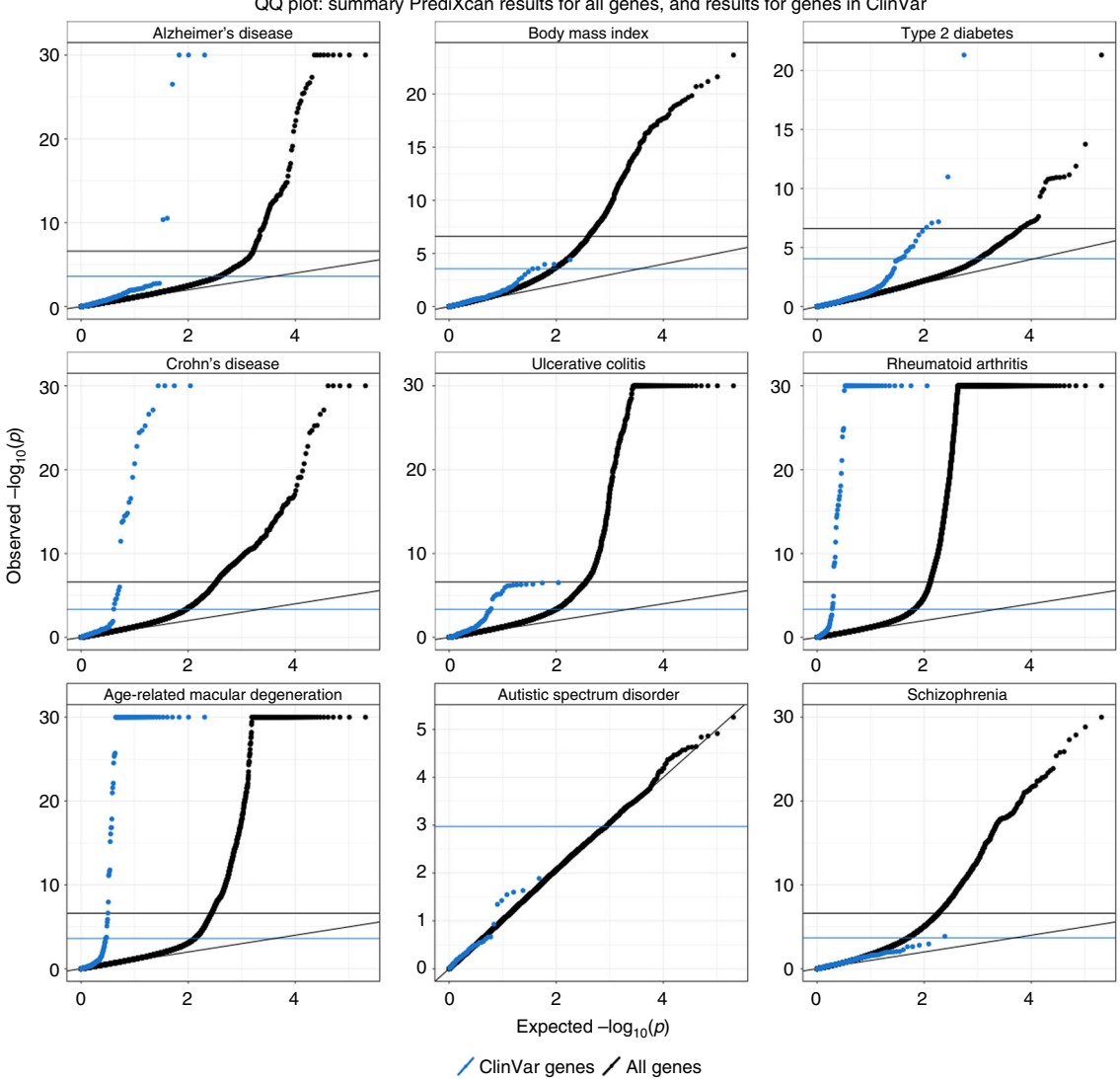

**Fig. 7** ClinVar genes show significant S-PrediXcan associations. Genes implicated in ClinVar tended to be more significant in S-PrediXcan for most diseases tested, except for schizophrenia and autism. This suggests that more moderate alteration of monogenic disease genes may contribute in a continuum of more moderate but related phenotypes. Alternatively, a more complex interplay between common and rare variation could be taking place such as higher tolerance to loss of function mutations in lower expressing haplotypes which could induce association with predicted expression. Blue circles correspond to the QQ plot of genes in ClinVar that were annotated with the phenotype and black circles correspond to all genes

of action. *C4A* is a gene with strong evidence of causal effect on schizophrenia risk via excessive synaptic pruning in the brain during development[32]. Our results show that *C4A* is associated with schizophrenia risk in all tissues ($p < 2.5 \times 10^{-7}$ in 36 tissue models and $p < 0.05$ for the remaining four tissue models).

*PCSK9* is a target of several LDL-C lowering drugs currently under trial to reduce cardiovascular events[34]. The STARNET study[35] profiled gene expression levels in cardiometabolic disease patients and showed tag SNP rs12740374 to be a strong eQTL for *PCSK9* in visceral fat but not in liver. Consistent with this, our S-PrediXcan results also show a highly significant association between *PCSK9* and LDL-C ($p \approx 10^{-13}$) in visceral fat and not in liver (our training algorithm did not yield a prediction model for *PCSK9*, i.e., there was no evidence of regulatory activity). In our results, however, the statistical evidence is much stronger in tibial nerve ($p \approx 10^{-27}$). Accordingly, in our training set (GTEx), there is much stronger evidence of regulation of this gene in tibial nerve compared to visceral fat.

Most associations highlighted here have high colocalization probabilities. See Supplementary Tables 3, 4, and 5. However, visceral fat association shows evidence of non colocalization (probability of independent signals P3 = 0.69 in LDL-C). It is possible that the relevant regulatory activity in visceral adipose tissue was not detected in the GTEx samples for various reasons but it was detected in tibial nerve. Thus by looking into all tissues' results we increase the window of opportunities where we can detect the association.

*PCSK9* yields colocalized signals for LDL-C levels in Tibial Nerve, Lung, and Whole blood. *SORT1* shows colocalization with LDL-C in liver (P4 ≈ 1) and pancreas (P4 = 0.90). *C4A* is colocalized with schizophrenia risk for the majority of the tissues (29/40) with a median colocalization probability of 0.82.

These examples demonstrate the power of studying regulation in a broad set of tissues and contexts and emphasize the challenges of determining causal tissues of complex traits based on in-silico analysis alone. Based on these results, we recommend to scan all tissues' models to increase the chances to detect the

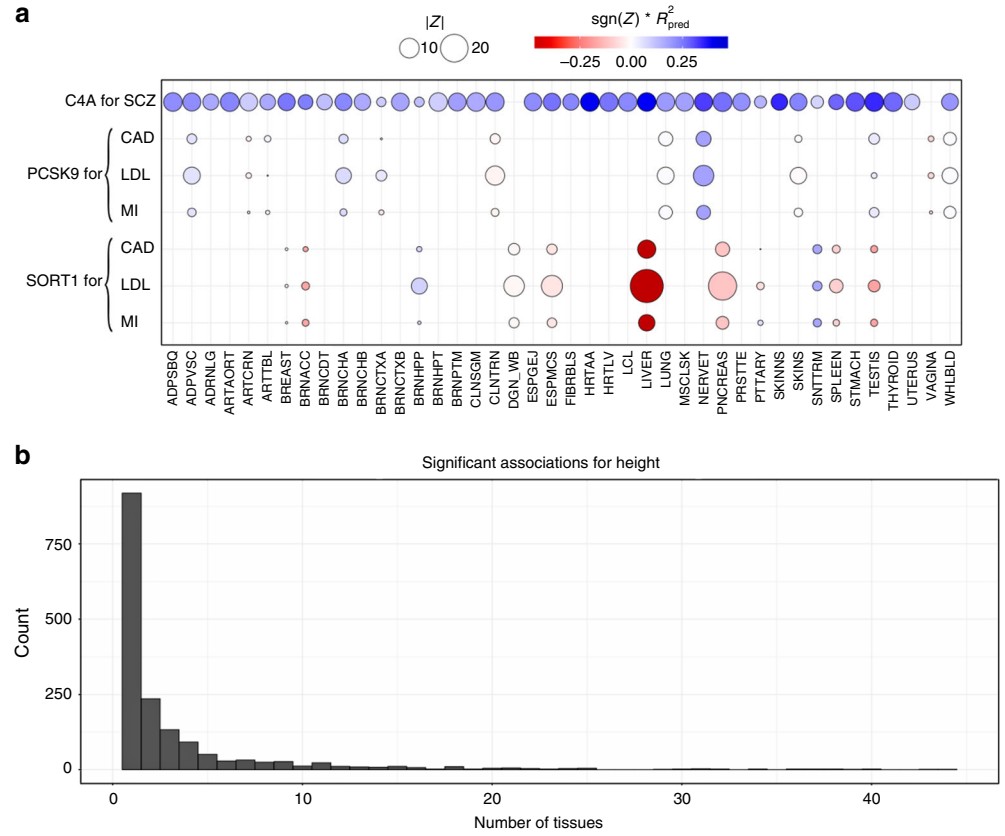

**Fig. 8** S-PrediXcan associations in different tissues. **a** Displays associations for *PCSK9*, *SORT1*, and *C4A* on relevant traits by tissue. This figure shows the association strength between three well studied genes and corresponding phenotypes. *C4A* associations with schizophrenia (SCZ) are significant across most tissues. *SORT1* associations with LDL-C, coronary artery disease (CAD), and myocardial infarction (MI) are most significant in liver. *PCSK9* associations with LDL-C, coronary artery disease (CAD), and myocardial infarction (MI) are most significant in tibial nerve. The size of the points represent the significance of the association between predicted expression and the traits indicated on the top labels. Red indicates negative correlation whereas blue indicates positive correlation. $R^2_{\mathrm{pred}}$ is a performance measure computed as the correlation squared between observed and predicted expression, cross validated in the training set. Darker points indicate larger genetic component and consequently more active regulation in the tissue. **b** Displays a histogram of the number of tissues for which a gene is significantly associated with height (other phenotypes show a similar pattern). Tissue abbreviations: ADPSBQ Adipose-Subcutaneous, ADPVSC Adipose-Visceral(Omentum), ADRNLG Adrenal Gland, ARTAORT Artery-Aorta, ARTCRN Artery-Coronary, ARTTBL Artery-Tibial, BLDDER Bladder, BRNAMY Brain-Amygdala, BRNACC Brain-Anterior cingulate cortex (BA24), BRNCDT Brain-Caudate(basal ganglia), BRNCHB Brain-Cerebellar Hemisphere, BRNCHA Brain-Cerebellum, BRNCTXA Brain-Cortex, BRNCTXB Brain-Frontal Cortex (BA9), BRNHPP Brain-Hippocampus, BRNHPT Brain-Hypothalamus, BRNNCC Brain Nucleus accumbens(basal ganglia), BRNPTM Brain-Putamen (basal ganglia), BRNSPC Brain-Spinal cord(cervical c-1), BRNSNG Brain-Substantia nigra, BREAST Breast-Mammary Tissue, LCL Cells-EBV-transformed lymphocytes, FIBRBLS Cells-Transformed fibroblasts, CVXECT Cervix-Ectocervix, CVSEND Cervix-Endocervix, CLNSGM Colon-Sigmoid, CLNTRN Colon-Transverse, ESPGEJ Esophagus-Gastroesophageal Junction, ESPMCS Esophagus-Mucosa, ESPMSL Esophagus-Muscularis, FLLPNT Fallopian Tube, HRTAA Heart-Atrial Appendage, HRTLV Heart-Left Ventricle, KDNCTX Kidney-Cortex, LIVER Liver, LUNG Lung, SLVRYG Minor Salivary Gland, MSCLSK Muscle-Skeletal, NERVET Nerve-Tibial, OVARY Ovary, PNCREAS Pancreas, PTTARY Pituitary, PRSTTE Prostate, SKINNS Skin-Not Sun Exposed (Suprapubic), SKINS Skin-Sun Exposed (Lower leg), SNTTRM Small Intestine-Terminal Ileum, SPLEEN Spleen, STMACH Stomach, TESTIS Testis, THYROID Thyroid, UTERUS Uterus, VAGINA Vagina, WHLBLD Whole Blood

relevant regulatory mechanism that mediates the phenotypic association. False positives can be controlled by Bonferroni correcting for the additional tests.

**Replication in an independent cohort**. We used data from the Resource for Genetic Epidemiology Research on Adult Health and Aging study (GERA, phs000674.v1.p1)[36,37]. This is a study led by the Kaiser Permanente Research Program on Genes, Environment, and Health (RPGEH) and the UCSF Institute for Human Genetics with over 100,000 participants. We downloaded the data from dbGaP and performed GWAS followed by S-PrediXcan analysis of 22 conditions available in the European subset of the cohort.

For replication, we chose Coronary Artery Disease (CAD), LDL cholesterol levels, Triglyceride levels, and schizophrenia, which had closely related phenotypes in the GERA study and had a sufficiently large number of Bonferroni significant associations in the discovery set. Analysis and replication of the type 2 diabetes phenotype can be found in ref.[38]. Coronary artery disease hits were compared with "Any cardiac event," LDL cholesterol and triglyceride level signals were compared with "Dyslipidemia," and schizophrenia was compared to "Any psychiatric event" in GERA.

High concordance between discovery and replication is shown in Fig. 9 where dyslipidemia association Z-scores are compared to LDL cholesterol Z-scores. The majority of gene-tissue pairs (92%, among the ones with Z-score magnitude greater than 2 in both

sets) have concordant direction of effects in the discovery and replication sets. The high level of concordance is supportive of an omnigenic trait architecture[39].

Following standard practice in meta-analysis, we consider a gene to be replicated when the following three conditions are met: the p-value in the replication set is <0.05, the direction of discovery and replication effects are the same, and the meta analyzed p-value is Bonferroni significant with the discovery threshold.

We display summary statistics for this replication analysis in Table 1. Among the 56 genes significantly associated with CAD in the discovery set, 6 (11%) were significantly associated with "Any cardiac event" in GERA. Using "Dyslipidemia" as the closest matching phenotype, 78.5% and 43.5% of LDL and triglyceride genes replicated, respectively. Among the 285 genes associated with schizophrenia in the discovery set, 51 (21%) replicated. The low replication rate for CAD and Schizophrenia is likely due to the broad phenotype definitions in the replication.

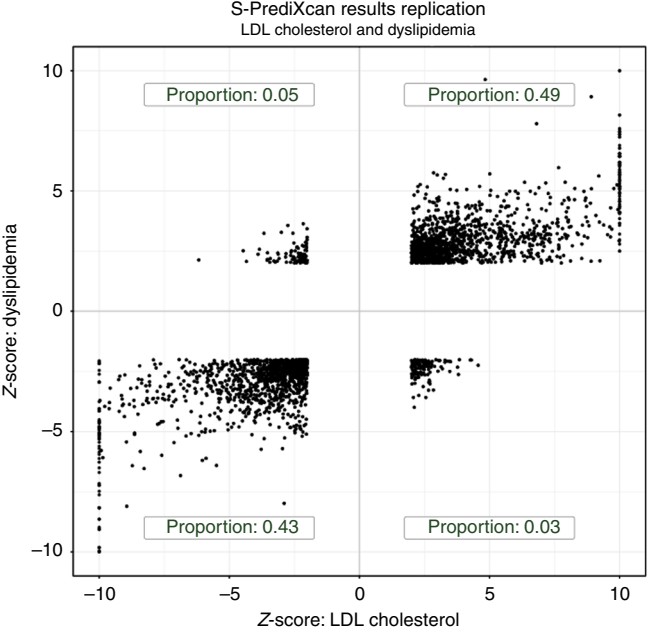

**Fig. 9** Discovery and replication Z-scores for lipid trait. This figure shows the Z-scores of the association between dyslipidemia (GERA) and predicted gene expression levels on the vertical axis and the Z-scores for LDL cholesterol on the horizontal axis. To facilitate visualization, very large Z-scores where thresholded to 10. Proportions in each quadrant were computed excluding Z-scores with magnitude smaller than 2 to filter out noise

We found no consistent replication pattern difference between colocalized and non-colocalized genes.

This is not unexpected if the LD pattern is similar between discovery and replication sets.

The full list of significant genes can be queried in gene2pheno. org.

## Discussion

Here we derive a mathematical expression to compute PrediXcan results without using individual level data, which greatly expands its applicability and is robust to study and reference set mismatches. This has not been done before. TWAS, which for the individual level approach only differs from PrediXcan on the prediction model used in the implementation, has been extended to use summary level data. When Gaussian imputation is used, the relationship between individual level and summary versions of TWAS is clear. This is not the case when extended to general weights (such as BSLMM). Our mathematical derivation shows the analytic difference between them explicitly.

We also add a post filtering step, to mitigate issues with LD-contamination. Based on consistency with PrediXcan and interpretability of results, we have chosen to use COLOC for filtering. COLOC has the limitation of assuming a single causal variant, and has reduced power in the presence of multiple causal variants. However, colocalization estimation is an active area of research and improved versions or methods will be adopted in the future. We find that BSLMM-based TWAS results have a larger proportion of non-colocalized genes as estimated by COLOC. This could be due to the single variant assumption in COLOC but we believe this is rather a consequence of the polygenic component of BSLMM predictors. Given the predominantly sparse architecture of gene expression traits[27], we believe that adding a polygenic component unnecessarily increases the exposure to LD-contamination.

Despite the generally good concordance between the summary and individual level methods, there were a handful of false positive results with S-PrediXcan much more significant than PrediXcan. This underscores the need to use closely matched LD information whenever possible.

We applied our framework to over 100 phenotypes using transcriptome prediction models trained in 44 tissues from the GTEx Consortium and generated a catalog of downstream phenotypic association results of gene expression variation, a growing resource for the community.

The enrichment of monogenic disease genes among related phenotype associations suggests that moderate alteration of expression levels as affected by common genetic variation may cause a continuum of phenotypic changes. Alternatively, a more complex interplay between common and rare variation could be taking place such as higher tolerance to loss of function mutations

**Table 1 Replication of results in GERA**

| Discovery phenotype | Replication phenotype | # Signif genes in disc set | # Replicated genes | $\pi_1$(all) in repl | $\pi_1$(sig) in repl | % Replicated genes | # Replicated coloc or underm |
|---|---|---|---|---|---|---|---|
| Coronary artery disease | Any cardiac event | 56 | 6 | 0.4% | 49.1% | 10.7% | 6 |
| LDL cholesterol | Dyslipidemia | 282 | 219 | 5.8% | 90.8% | 78.5% | 184 |
| Triglycerides | Dyslipidemia | 233 | 100 | 5.8% | 73.1% | 43.5% | 69 |
| Schizophrenia | Any psychiatric event | 285 | 60 | 1.2% | 47.6% | 21.1% | 51 |

Significant genes/tissue pairs were replicated using a closely matched phenotype in an independent dataset from the GERA cohort[36]. The criteria consisted in significance threshold for replication at p < 0.05, concordant directions of effect, and meta analysis p-value less than the Bonferroni threshold in the discovery set. $\pi_1$ is an estimate of proportion of true positives in the replication set. $\pi_1$(all) uses all gene-tissue pairs whereas $\pi_1$(sig) is computed using only gene-tissue pairs that were significant in the discovery set. The column '# replicated genes coloc or underm' is the number of replicated genes excluding the ones for which there was strong evidence of independent GWAS and eQTL signals

in lower expressing haplotypes which could induce association with predicted expression[40].

We are finding that most trait associations are tissue specific; i.e., they are detected in a handful of tissues. However, we also find that expected tissues given known biology do not necessarily rank among the top enriched tissues. This suggests context specificity of the pathogenic mechanism; specific developmental stage or environmental conditions may be necessary to detect the regulatory event. On the other hand, we are detecting associations in unexpected tissues which suggests a sharing of regulation across multiple tissues/contexts or perhaps novel biology that takes place in these tissues. In either case, agnostic scanning of a broad set of tissues is necessary to discover these mechanisms.

## Methods

**Summary-PrediXcan formula**. Figure 1b shows the main analytic expression used by Summary-PrediXcan for the Z-score (Wald statistic) of the association between predicted gene expression and a phenotype. The input variables are the weights used to predict the expression of a given gene, the variance and covariances of the markers included in the prediction, and the GWAS coefficient for each marker. The last factor in the formula can be computed exactly in principle, but we would need additional information that is unavailable in typical GWAS summary statistics output such as phenotype variance and sample size. Dropping this factor from the formula does not affect the accuracy of the results as demonstrated in the close to perfect concordance between PrediXcan and Summary-PrediXcan results on the diagonal of Fig. 2a.

The approximate formula we use is:

$$Z_g \approx \sum_{l \in \text{Model}_g} w_{lg} \frac{\widehat{\sigma_l}}{\widehat{\sigma_g}} \frac{\hat{\beta}_l}{\text{se}(\hat{\beta}l)}, \tag{1}$$

where $w_{lg}$ is the weight of SNP $l$ in the prediction of the expression of gene $g$; $\hat{\beta}_l$ is the GWAS regression coefficients for SNP $l$; $\text{se}(\hat{\beta}_l)$ is standard error of $\beta$, $\hat{\sigma}_l$ is the estimated variance of SNP $l$, and $\hat{\sigma}_g$ is the estimated variance of the predicted expression of gene $g$; and dosage and alternate allele are assumed to be the same.

The inputs are based, in general, on data from three different sources: study set (e.g., GWAS study set), expression training set (e.g., GTEx, DGN), population reference set (e.g., the training set or 1000 Genomes).

The study set is the main dataset of interest from which the genotype and phenotypes of interest are gathered. The regression coefficients and standard errors are computed based on individual-level data from the study set or a SNP-level meta-analysis of multiple GWAS. Training sets are the reference transcriptome datasets used for the training of the prediction models (GTEx, DGN, Framingham, etc.) thus the weights $w_{lg}$ are computed from this set. Training sets can also be used to generate variance and covariances of genetic markers, which will usually be different from the study sets. When individual level data are not available from the training set we use population reference sets such as 1000 Genomes data. In the most common use scenario, users will need to provide only GWAS results using their study set. The remaining parameters are pre-computed, and published in PredictDB.

**Association enrichment**. We display the enrichment for selected phenotypes in Supplementary Fig. 5, measured as mean($Z^2$). For visualization purposes, we selected 25 phenotypes from different categories such as anthropometric traits, cardiometabolic traits, autoimmune diseases, and psychiatric conditions (please see figure caption for the list of selected phenotypes). The simple mean of $Z^2$ for all gene-tissue pairs in a phenotype was taken.

**Derivation of summary-PrediXcan formula**. The goal of summary-PrediXcan is to infer the results of PrediXcan using only GWAS summary statistics. Individual level data are not needed for this algorithm. We will introduce some notations for the derivation of the analytic expressions of S-PrediXcan.

**Notation and preliminaries**. $Y$ is the $n$-dimensional vector of phenotype for individuals $i = 1, n$. $X_l$ is the allelic dosage for SNP $l$. $T_g$ is the predicted expression (or estimated GREx, genetically regulated expression). $w_{lg}$ are weights to predict expression $T_g = \sum_{l \in \text{Model } g} w_{lg} X_l$, derived from an independent training set.

We model the phenotype as linear functions of $X_l$ and $T_g$

$$Y = \alpha_1 + X_l \beta_l + \eta$$

$$Y = \alpha_2 + T_g \gamma_g + \epsilon$$

where $\alpha_1$ and $\alpha_2$ are intercepts, $\eta$ and $\epsilon$ error terms independent of $X_l$ and $T_g$, respectively. Let $\hat{\gamma}_g$ and $\hat{\beta}_l$ be the estimated regression coefficients of $Y$ regressed on

$T_g$ and $X_l$, respectively. $\hat{\gamma}_g$ is the result (effect size for gene $g$) we get from PrediXcan whereas $\hat{\beta}_l$ is the result from a GWAS for SNP $l$.

We will denote as $\widehat{\text{Var}}$ and $\widehat{\text{Cov}}$ the operators that compute the sample variance and covariance, i.e.,: $\widehat{\text{Var}}(Y) = \hat{\sigma}_Y^2 = \sum_{i=1,n} (Y_i - \overline{Y})^2/(n-1)$ with $\overline{Y} = \sum_{i=1,n} Y_i/n$. Let $\hat{\sigma}_l^2 = \widehat{\text{Var}}(X_l)$, $\hat{\sigma}_g^2 = \widehat{\text{Var}}(T_g)$ and $\Gamma_g = (\mathbf{X} - \overline{\mathbf{X}})'(\mathbf{X} - \overline{\mathbf{X}})/n$, where $\mathbf{X}'$ is the $p \times n$ matrix of SNP data and $\overline{\mathbf{X}}$ is a $n \times p$ matrix where column $l$ has the column mean of $X_l$ ($p$ being the number of SNPs in the model for gene $g$, typically $p \ll n$).

With this notation, our goal is to infer PrediXcan results ($\hat{\gamma}_g$ and its standard error) using only GWAS results ($\hat{\beta}_l$ and their standard error), estimated variances of SNPs ($\hat{\sigma}_l^2$), estimated covariances between SNPs in each gene model ($\Gamma_g$), and prediction model weights $w_{lg}$.

**Input**: $\hat{\beta}_l$, $\text{se}(\hat{\beta}_l)$, $\hat{\sigma}_l^2$, $\Gamma_g$, $w_{lg}$. **Output**: $\hat{\gamma}_g/\text{se}(\hat{\gamma}_g)$.

Next we list the properties and definitions used in the derivation

$$\hat{\gamma}_g = \frac{\widehat{\text{Cov}}(T_g, Y)}{\widehat{\text{Var}}(T_g)} = \frac{\widehat{\text{Cov}}(T_g, Y)}{\hat{\sigma}_g^2} \tag{2}$$

and

$$\hat{\beta}_l = \frac{\widehat{\text{Cov}}(X_l, Y)}{\widehat{\text{Var}}(X_l)} = \frac{\widehat{\text{Cov}}(X_l, Y)}{\hat{\sigma}_l^2} \tag{3}$$

The proportion of variance explained by the covariate ($T_g$ or $X_l$) can be expressed as

$$R_g^2 = \hat{\gamma}_g^2 \frac{\hat{\sigma}_g^2}{\hat{\sigma}_Y^2}$$

$$R_l^2 = \hat{\gamma}_l^2 \frac{\hat{\sigma}_l^2}{\hat{\sigma}_Y^2}$$

By definition

$$T_g = \sum_{l \in \text{Model}_g} w_{lg} X_l$$

Thus $\widehat{\text{Var}}(T_g) = \hat{\sigma}_g^2$ can be computed as

$$\begin{aligned} \hat{\sigma}_g^2 &= \widehat{\text{Var}}\left(\sum_{l \in \text{Model}_g} w_{lg} X_l\right) \\ &= \widehat{\text{Var}}(\mathbf{W}_g \mathbf{X}_g) \\ &= \mathbf{W}_g' \widehat{\text{Var}}(\mathbf{X}_g) \mathbf{W}_g \end{aligned}$$

where $\mathbf{W}_g$ is the vector of $w_{lg}$ for SNPs in the model of $g$. By definition, $\Gamma_g$ is $\widehat{\text{Var}}(\mathbf{X}_g)$, the sample covariance of $\mathbf{X}_g$, so that we arrive to

$$\hat{\sigma}_g^2 = \mathbf{W}_g' \Gamma \mathbf{W}_g \tag{4}$$

Calculation of regression coefficient $\hat{\gamma}_g$
$\hat{\gamma}_g$ can be expressed as

$$\begin{aligned} \hat{\gamma}_g &= \frac{\widehat{\text{Cov}}(T_g, Y)}{\hat{\sigma}_g^2} \\ &= \frac{\widehat{\text{Cov}}\left(\sum_{l \in \text{Model}_g} w_{lg} X_l, Y\right)}{\hat{\sigma}_g^2} \\ &= \sum_{l \in \text{Model}_g} \frac{w_{lg} \widehat{\text{Cov}}(X_l, Y)}{\hat{\sigma}_g^2} \end{aligned}$$

where we used the linearity of $\widehat{\text{Cov}}$ in the last step. Using Eq. (3), we arrive to

$$\hat{\gamma}_g = \sum_{l \in \text{Model}_g} \frac{w_{lg} \hat{\beta}_l \hat{\sigma}_l^2}{\hat{\sigma}_g^2} \tag{5}$$

Calculation of standard error of $\hat{\gamma}_g$
Also from the properties of linear regression we know that

$$\text{se}^2(\hat{\gamma}_g) = \text{Var}(\hat{\gamma}_g) = \frac{\hat{\sigma}_\epsilon^2}{n \hat{\sigma}_g^2} = \frac{\hat{\sigma}_Y^2 (1 - R_g^2)}{n \hat{\sigma}_g^2} \tag{6}$$

In this equation, $\hat{\sigma}_Y^2/n$ is not necessarily known but can be estimated using the equation analogous to (6) for $\beta_l$

$$\text{se}^2\left(\hat{\beta}_l\right) = \frac{\hat{\sigma}_Y^2(1 - R_l^2)}{n\,\hat{\sigma}_l^2} \qquad (7)$$

Thus:

$$\frac{\hat{\sigma}_Y^2}{n} = \frac{\text{se}^2\left(\hat{\beta}_l\right)\hat{\sigma}_l^2}{(1 - R_l^2)} \qquad (8)$$

Notice that the right hand side of (8) is dependent on the SNP $l$ while the left hand side is not. This equality will hold only approximately in our implementation since we will be using approximate values for $\hat{\sigma}_l^2$, i.e., from reference population, not the actual study population.

**Calculation of Z-score.** To assess the significance of the association, we need to compute the ratio of the estimated effect size $\hat{\gamma}_g$ and standard error $\text{se}(\hat{\gamma}_g)$, or Z-score,

$$Z_g = \frac{\hat{\gamma}_g}{\text{se}(\hat{\gamma}_g)} \qquad (9)$$

with which we can compute the p-value as $p = 2\Phi(-|Z_g|)$ where $\Phi(.)$ is the normal CDF function. Thus

$$
\begin{aligned}
Z_g &= \frac{\hat{\gamma}_g}{\text{se}(\hat{\gamma}_g)} \\
&= \sum_{l \in \text{Model}_g} \frac{w_{lg}\hat{\beta}_l\hat{\sigma}_l^2}{\hat{\sigma}_g^2}\sqrt{\frac{n}{\hat{\sigma}_Y^2}\frac{\hat{\sigma}_g^2}{(1-R_g^2)}} \\
&= \sum_{l \in \text{Model}_g} \frac{w_{lg}\hat{\beta}_l\hat{\sigma}_l^2}{\hat{\sigma}_g^2}\sqrt{\frac{(1-R_l^2)}{\text{se}^2\left(\hat{\beta}_l\right)\hat{\sigma}_l^2}\frac{\hat{\sigma}_g^2}{(1-R_g^2)}}
\end{aligned}
$$

where we used Eqs. (5) and (6) in the second line and Eq. (8) in the last step. So

$$Z_g = \sum_{l \in \text{Model}_g} w_{lg}\frac{\hat{\sigma}_l}{\hat{\sigma}_g}\frac{\hat{\beta}_l}{\text{se}\left(\hat{\beta}_l\right)}\sqrt{\frac{(1-R_l^2)}{(1-R_g^2)}} \qquad (10)$$

$$\approx \sum_{l \in \text{Model}_g} w_{lg}\frac{\hat{\sigma}_l}{\hat{\sigma}_g}\frac{\hat{\beta}_l}{\text{se}\left(\hat{\beta}_l\right)} \qquad (11)$$

Based on results with actual and simulated data for realistic effect size ranges, we have found that the last approximation does not affect our ability to identify the association. The approximation becomes inaccurate only when the effect sizes are very large. But in these cases, the small decrease in statistical efficiency induced by the approximation is compensated by the large power to detect the larger effect sizes.

**Calculation of $\sigma_g$ in reference set.** The variance of predicted expression is computed using Eq. (4) which takes weights for each SNP in the prediction model and the correlation (LD) between the SNPs. The correlation is computed in a reference set such as 1000G or in the training set.

**Expression model training.** To train our prediction models, we obtained genotype data and normalized gene expression data collected by the GTEx Project. We used 44 different tissues sampled by GTEx and thus generated 44 different tissue-wide models (dbGaP Accession phs000424.v6.p1). Sample sizes for different tissues range from 70 (Uterus) to 361 (Muscle—Skeletal). The models referenced in this paper make use of the GTEx Project's V6p data, a patch to the version 6 data and makes use of improved gene-level annotation. We removed ambiguously stranded SNPs from genotype data, i.e., ref/alt pairs A/T, C/G, T/A, G/C. Genotype data was filtered to include only SNPs with MAF > 0.01. For each tissue, normalized gene expression data was adjusted for covariates such as gender, sequencing platform, the top three principal components from genotype data and top PEER Factors. The number of PEER Factors used was determined by sample size: 15 for $n < 150$, 30 for $n$ between 150 and 250, and 35 for $n > 250$. Covariate data was provided by GTEx. For our analysis, we used protein-coding genes only.

For each gene-tissue pair for which we had adjusted expression data, we fit an Elastic-Net model based on the genotypes of the samples for the SNPs located within 1 Mb upstream of the gene's transcription start site and 1 Mb downstream of the transcription end site. We used the R package glmnet with mixing parameter alpha equal to 0.5, and the penalty parameter lambda was chosen through 10-fold

**Cross-validation.** Once we fit all models, we retained only those with q-value less than 0.05[41] For each tissue examined, we created a sqlite database to store the weights of the prediction models, as well as other statistics regarding model training. Supplementary Table 6 contains summary statistics on the models for each GTEx tissue. These databases have been made available for download at PredictDB.org.

**Online Catalog and SMR, COLOC, TWAS.** Supplementary Data 2 shows the list of GWA/GWAMA studies we considered in this analysis. We applied S-PrediXcan to these studies using the transcriptome models trained on GTEx studies for patched version 6. For simplicity, S-PrediXcan only considers those SNPs that have a matching set of alleles in the prediction model, and adjusts the dosages (2 dosage) if the alleles are swapped.

To make the results of this study broadly accessible, we built a Postgre SQL relational database to store S-PrediXcan results, and serve them via a web application http://gene2pheno.org.

We also applied SMR[16] to the same set of GWAMA studies, using the GTEx eQTL associations. We downloaded version 0.66 of the software from the SMR website, and ran it using the default parameters. We converted the GWAMA and GTEx eQTL studies to SMR input formats. In order to have SMR compute the colocalization test, for those few GWAMA studies where allele frequency was not reported, we filled in with frequencies from the 1000 Genomes Project[42] as an approximation. We also used the 1000 Genomes genotype data as reference panel for SMR.

Next we ran COLOC[18] (as downloaded from the Comprehensive R Archive Network) over the same set of GWAMA and eQTL studies. We used the Approximate Bayes Factor colocalization analysis, with effect sizes, their standard errors, allele frequencies and sample sizes as arguments. When the frequency information was missing from the GWAS, we filled in with data from the 1000 Genomes Project.

For comparison purposes, we have also included the results of the application of Summary-TWAS to 30 traits publicly shared by the authors[24].

**Comparison with TWAS.** Formal similarity with TWAS can be made more explicit by rewriting S-PrediXcan formula in matrix form. With the following notation and definitions

$$
\begin{aligned}
\tilde{\mathbf{W}}_g &= \left(\sigma_1 w_{1g}, \dots, \sigma_p w_{pg}\right)' \\
\mathbf{Z}_{\text{SNPs}} &= \left(Z_1, \dots, Z_p\right)' \\
&= \left(\frac{\hat{\beta}_1}{\text{se}(\hat{\beta}_1)}, \dots, \frac{\hat{\beta}_p}{\text{se}(\hat{\beta}_p)}\right)'
\end{aligned}
$$

and correlation matrix of SNPs in the model for gene $g$

$$\Sigma_g = \text{diag}\left(\frac{1}{\hat{\sigma}_1}, \dots, \frac{1}{\hat{\sigma}_p}\right)\cdot\Gamma_g\cdot\text{diag}\left(\frac{1}{\hat{\sigma}_1}, \dots, \frac{1}{\hat{\sigma}_p}\right)$$

it is quite straightforward to write the numerator in (1) and (11) as $\tilde{\mathbf{W}}_g\cdot\mathbf{Z}_{\text{SNPs}}$ and in the denominator, the variance of the predicted expression level of gene $g$, as

$$\tilde{\mathbf{W}}_g'\cdot\Sigma_g\cdot\tilde{\mathbf{W}}_g$$

Thus

$$Z_g = \frac{\tilde{\mathbf{W}}_g\cdot\mathbf{Z}_{\text{SNPs}}}{\sqrt{\tilde{\mathbf{W}}_g'\cdot\Sigma_g\cdot\tilde{\mathbf{W}}_g}}$$

This equation has the same form as the TWAS expression if we use the scaled weight vector $\tilde{\mathbf{W}}_g$ instead of $\mathbf{W}_g$. Summary-TWAS imputes the Z-score for the gene-level result assuming that under the null hypothesis, the Z-scores are normally distributed with the same correlation structure as the SNPs; whereas in S-PrediXcan we compute the results of PrediXcan using summary statistics. Thus, S-TWAS and S-PrediXcan yield equivalent mathematical expressions (after setting the factor $\sqrt{\frac{(1-R_l^2)}{(1-R_g^2)}} \approx 1$).

**Summary-PrediXcan with only top eQTL as predictor.** The S-PrediXcan formula when only the top eQTL is used to predict the expression level of a gene can be expressed as

$$
\begin{aligned}
Z_{S-\text{PrediXcan}} &= \sum_{l \in \text{Model}_g} w_{lg}\frac{\hat{\sigma}_l}{\hat{\sigma}_g}\frac{\hat{\beta}_l}{\text{se}(\hat{\beta}_l)} \\
&= w_{1g}\frac{\hat{\sigma}_1}{\sqrt{w_{1g}^2\hat{\sigma}_1^2}}Z_1 \\
&= Z_1
\end{aligned}
$$

where $Z_1$ is the GWAS Z-score of the top eQTL in the model for gene. Thus

$$Z^2_{\text{top eQTL S−PrediXcan}} = Z^2_{\text{GWAS}} \qquad (12)$$

**Comparison with SMR**. SMR quantifies the strength of the association between expression levels of a gene and complex traits with $T_{\text{SMR}}$ using the following function of the eQTL and GWAS Z-score statistics

$$T_{\text{SMR}} = \frac{Z^2_{\text{eQTL}} Z^2_{\text{GWAS}}}{Z^2_{\text{eQTL}} + Z^2_{\text{GWAS}}} \qquad (13)$$

Here $Z_{\text{eQTL}}$ is the Z-score (=effect size/standard error) of the association between SNP and gene expression, and $Z_{\text{GWAS}}$ is the Z-score of the association between SNP and trait.

This SMR statistic ($T_{\text{SMR}}$) is not a $\chi^2_1$ random variable as assumed in ref. [16]. To prove this, we performed simulations following those described in ref. [16]. We generated $10^5$ pairs of values for $Z^2_{\text{GWAS}}$ and $Z^2_{\text{eQTL}}$. $Z^2_{\text{GWAS}}$ was sampled from a $\chi^2_1$ distribution. $Z^2_{\text{eQTL}}$ was sampled from a non-central $\chi^2_1$ distribution with parameter $\lambda = 29$ (a value chosen to mimic results from[29], see ref. [16]). Only pairs with eQTLs satisfying genome-wide significance ($p < 5 \times 10^{-8}$) were kept. We performed a QQ plot and observed deflation when comparing to random values sampled from a $\chi^2_1$ distribution (Fig. 5e). This simulation was repeated 1000 times, and we observed a mean of $T_{\text{SMR}}$ close to 0.93 (Fig. 5f).

Only in two extreme cases, the chi-square approximation holds, when $Z_{\text{eQTL}} \gg Z_{\text{GWAS}}$ or $Z_{\text{eQTL}} \ll Z_{\text{GWAS}}$. In these extremes, we can apply Taylor expansions to find an interpretable form of the SMR statistic.

If $Z_{\text{eQTL}} \gg Z_{\text{GWAS}}$, i.e., the eQTL association is much more significant than the GWAS association,

$$T_{\text{SMR}} = \frac{Z^2_{\text{GWAS}}}{1 + \frac{Z^2_{\text{GWAS}}}{Z^2_{\text{eQTL}}}} \approx Z^2_{\text{GWAS}} \left(1 - \frac{Z^2_{\text{GWAS}}}{Z^2_{\text{eQTL}}}\right) \qquad (14)$$

so that for large enough $Z^2_{\text{eQTL}}$ relative to $Z^2_{\text{GWAS}}$,

$$T_{\text{SMR}} \approx Z^2_{\text{GWAS}} = Z^2_{\text{top eQTL S−PrediXcan}} \qquad (15)$$

using Eq. 12. Thus, in this case, the SMR statistic is slightly smaller than the (top eQTL based) S-PrediXcan $\chi_1$-square. This reduced significance is accounting for the uncertainty in the eQTL association. As the evidence for eQTL association grows, the denominator $Z^2_{\text{eQTL}}$ increases and the difference tends to 0.

On the other extreme when the GWAS association is much stronger than the eQTLs, $Z_{\text{eQTL}} \ll Z_{\text{GWAS}}$,

$$T_{\text{SMR}} = \frac{Z^2_{\text{eQTL}}}{1 + \frac{Z^2_{\text{eQTL}}}{Z^2_{\text{GWAS}}}} \approx Z^2_{\text{eQTL}} \left(1 - \frac{Z^2_{\text{eQTL}}}{Z^2_{\text{GWAS}}}\right) \qquad (16)$$

so that analogously

$$T_{\text{SMR}} \approx Z^2_{\text{eQTL}} \qquad (17)$$

In both extremes, the SMR statistic significance is approximately equal to the less significant of the two statistics GWAS or eQTL, albeit strictly smaller.

In between the two extremes, the right distribution must be computed using numerical methods. When we look at the empirical distribution of the SMR statistic's p-value against the GWAS and eQTL (top eQTL for the gene) p-values, we find the ceiling of the SMR statistic is maintained as shown in Fig. 5e, f.

**GERA imputation**. Genotype files were obtained from dbGaP, and updated to release 35 of the probe annotations published by Affymetrix via PLINK[43]. Probes were filtered out that had a minor allele frequency of <0.01, were missing in >10% of subjects, or did not fit Hardy-Weinberg equilibrium. Subjects were dropped that had an unexpected level of heterozygosity ($F > 0.05$). Finally the HRC-1000G-check-bim.pl script (http://www.well.ox.ac.uk/~wrayner/tools/) was used to perform some final filtering and split data by chromosome. Phasing (via eagle v2.3[44]) and imputation against the HRC r1.1 2016 panel[45] (via minimac3) were carried out by the Michigan Imputation Server[46].

**GERA GWAS and MetaXcan Application**. European samples had been split into ten groups during imputation to ease the computational burden on the Michigan server, so after obtaining the imputed .vcf files, we used the software PLINK[43] to convert the genotype files into the PLINK binary file format and merge the ten groups of samples together, while dropping any variants not found in all sample groups. For the association analysis, we performed a logistic regression using PLINK, and following QC practices from ref. [14] we filtered out individuals with genotype missingness >0.03 and filtered out variants with minor allele frequency <0.01, missingness >0.05, out of Hardy-Weinberg equilibrium significant at 1e-6, or had imputation quality <0.8. We used gender and the first ten genetic principal components as obtained from dbGaP as covariates. Following all filtering, our analysis included 61,444 European samples with 7,120,064 variants. MetaXcan was then applied to these GWAS results, using the 45 prediction models (GTEx and DGN).

**Code Availability**. We make our software publicly available on a GitHub repository: https://github.com/hakyimlab/MetaXcan. A short working example can be found on the GitHub page; more extensive documentation can be found on the project's wiki page.

**Data availability**. The underlying GWAS results used in this analysis were downloaded from publicly available resources listed in Supplementary Data 2. The relevant GTEx gene expression data was obtained from dbGAP using accession phs000424.v6.p1. The GERA study was downloaded from dbGAP using accession number phs000674.v2.p2. WTCCC data was downloaded from WTCCC EGA european genome-phenome archive.

The list of ClinVar genes was downloaded from https://www.ncbi.nlm.nih.gov/clinvar/. TWAS results published in ref. [24] were used. Prediction model weights and covariances for different tissues are available from the predictdb.org resource. The results of MetaXcan applied to the 44 human tissues and a broad set of phenotypes can be queried in gene2pheno.org, and we make the full data set of results available via the public GitHub repository https://github.com/hakyimlab/MetaXcan.

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

## Acknowledgements

We acknowledge the following US National Institutes of Health grants: R01MH107666 (H.K.I.), T32 MH020065 (K.P.S.), R01 MH101820 (GTEx), P30 DK020595 (Diabetes Research and Training Center), F31 DK101202 (J.M.T.), P50 DA037844 (Rat Genomics), and P50 MH094267 (Conte). H.E.W. was supported in part by start-up funds from the Loyola University Chicago.

The Genotype-Tissue Expression (GTEx) Project was supported by the Common Fund of the Office of the Director of the National Institutes of Health. Additional funds were provided by the NCI, NHGRI, NHLBI, NIDA, NIMH, and NINDS. Donors were enrolled at Biospecimen Source Sites funded by NCI SAIC-Frederick, Inc. (SAIC-F) subcontracts to the National Disease Research Interchange (10XS170), Roswell Park Cancer Institute (10XS171), and Science Care, Inc. (X10S172). The Laboratory, Data Analysis, and Coordinating Center (LDACC) was funded through a contract (HHSN268201000029C) to the Broad Institute, Inc. Biorepository operations were funded through an SAIC-F subcontract to the Van Andel Institute (10ST1035). Additional data repository and project management were provided by SAIC-F (HHSN261200800001E). The Brain Bank was supported by a supplements to the University of Miami grants DA006227 and DA033684 and to contract N01MH000028. Statistical Methods development grants were made to the University of Geneva (MH090941 and MH101814), the University of Chicago (MH090951, MH090937, MH101820, and MH101825), the University of North Carolina—Chapel Hill (MH090936 and MH101819), the Harvard University (MH090948), the Stanford University (MH101782), the Washington University St Louis (MH101810), and the University of Pennsylvania (MH101822). The data used for the analyses described in this manuscript were obtained from dbGaP accession number phs000424.v6.p1 on 06/17/2016.

This study makes use of data generated by the Wellcome Trust Case-Control Consortium. A full list of the investigators who contributed to the generation of the data is available from www.wtccc.org.uk. Funding for the project was provided by the Wellcome Trust under award 076113 and 085475.

This work was completed in part with resources provided by the University of Chicago Research Computing Center, Bionimbus[47], and the Center for Research Informatics. The Center for Research Informatics is funded by the Biological Sciences Division at the University of Chicago with additional funding provided by the Institute for Translational Medicine, CTSA grant number UL1 TR000430 from the National Institutes of Health.

## Author contributions

A.N.B. contributed to S-PrediXcan software developmentl; executed S-PredixCan runs on the GWAS traits; developed framework for comparing PrediXcan and S-PrediXcan in simulated, cellular and WTCCC phenotypes; ran COLOC and SMR; developed gene2-pheno.org database and web dashboard; contributed to the main text, supplement, figures, and analyses. S.P.D. performed the GTEx model training; ran the GERA GWAS; contributed to the main text. J.M.T. contributed to the main text. J.Z. contributed to the figures and predictdb.org resource. E.S.T. contributed to S-PrediXcan software. H.E.W. contributed to the main text and analysis. K.P.S. ran PrediXcan on WTCCC data and contributed to the analysis. R.B. contributed to the main text and figures. T.G. ran imputation of GERA genotypes. T.L.E. contributed to the analysis. L.M.H. contributed to the main text. E.A.S. contributed to the main text. D.L.N. contributed to the main text and analysis. N.J.C. contributed to the analysis. H.K.I. conceived the method, supervised the project, performed analysis, contributed to the main text, supplement, and figures. The GTEx consortium authors contributed in the collection, gathering, and processing of GTEx study data used for training transcriptome prediction models and running COLOC and SMR.

## Additional information

**Competing interests:** The authors declare no competing interests.

## GTEx Consortium

## Laboratory, Data Analysis & Coordinating Center (LDACC)—Analysis Working Group

François Aguet[10], Kristin G. Ardlie[10], Beryl B. Cummings[10,11], Ellen T. Gelfand[10], Gad Getz[10,12], Kane Hadley[10], Robert E. Handsaker[10,13], Katherine H. Huang[10], Seva Kashin[10,13], Konrad J. Karczewski[10,11], Monkol Lek[10,11], Xiao Li[10], Daniel G. MacArthur[10,11], Jared L. Nedzel[10], Duyen T. Nguyen[10], Michael S. Noble[10], Ayellet V. Segrè[10], Casandra A. Trowbridge[10] & Taru Tukiainen[10,11]

[10]The Broad Institute of Massachusetts Institute of Technology and Harvard University, Cambridge, MA 02142, USA. [11]Analytic and Translational Genetics Unit, Massachusetts General Hospital, Boston, MA 02114, USA. [12]Massachusetts General Hospital Cancer Center, Department of Pathology, Massachusetts General Hospital, Boston, MA 02114, USA. [13]Department of Genetics, Harvard Medical School, Boston, MA 02114, USA

## Statistical Methods groups—Analysis Working Group

Nathan S. Abell[14,15], Brunilda Balliu[15], Ruth Barshir[16], Omer Basha[16], Alexis Battle[17], Gireesh K. Bogu[18,19], Andrew Brown[20,21,22], Christopher D. Brown[23], Stephane E. Castel[24,25], Lin S. Chen[26], Colby Chiang[27], Donald F. Conrad[28,29], Farhan N. Damani[17], Joe R. Davis[14,15], Olivier Delaneau[20,21,22], Emmanouil T. Dermitzakis[20,21,22], Barbara E. Engelhardt[30], Eleazar Eskin[31,32], Pedro G. Ferreira[33,34], Laure Frésard[14,15], Eric R. Gamazon[35,36,37], Diego Garrido-Martín[18,19], Ariel D.H. Gewirtz[38], Genna Gliner[39], Michael J. Gloudemans[14,15,40], Roderic Guigo[18,19,41], Ira M. Hall[27,28,42], Buhm Han[43], Yuan He[44], Farhad Hormozdiari[31], Cedric Howald[20,21,22], Brian Jo[38], Eun Yong Kang[31], Yungil Kim[17], Sarah Kim-Hellmuth[24,25], Tuuli Lappalainen[24,25], Gen Li[45], Xin Li[15], Boxiang Liu[14,15,46], Serghei Mangul[31], Mark I. McCarthy[47,48,49], Ian C. McDowell[50], Pejman Mohammadi[24,25], Jean Monlong[18,19,51], Stephen B. Montgomery[14,15], Manuel Muñoz-Aguirre[18,19,52], Anne W. Ndungu[47], Andrew B. Nobel[53,54], Meritxell Oliva[55,56], Halit Ongen[20,21,22], John J. Palowitch[53], Nikolaos Panousis[20,21,22], Panagiotis Papasaikas[18,19], YoSon Park[23], Princy Parsana[17], Anthony J. Payne[47], Christine B. Peterson[57], Jie Quan[58], Ferran Reverter[18,19,59], Chiara Sabatti[60,61], Ashis Saha[17], Michael Sammeth[62], Alexandra J. Scott[27], Andrey A. Shabalin[63], Reza Sodaei[18,19], Matthew Stephens[64,65], Barbara E. Stranger[55,56,66], Benjamin J. Strober[44], Jae Hoon Sul[67], Emily K. Tsang[15,40], Sarah Urbut[65], Martijn van de Bunt[47,48], Gao Wang[65], Xiaoquan Wen[68], Fred A. Wright[69], Hualin S. Xi[58], Esti Yeger-Lotem[16,70], Zachary Zappala[14,15], Judith B. Zaugg[71] & Yi-Hui Zhou[69]

[14]Department of Genetics, Stanford University, Stanford, CA 94305, USA. [15]Department of Pathology, Stanford University, Stanford, CA 94305, USA. [16]Department of Clinical Biochemistry and Pharmacology, Faculty of Health Sciences, BenGurion University of the Negev, Beer-Sheva 84105, Israel. [17]Department of Computer Science, Johns Hopkins University, Baltimore, MD 21218, USA. [18]Centre for Genomic Regulation (CRG), The Barcelona Institute for Science and Technology, 08003 Barcelona, Spain. [19]Universitat Pompeu Fabra (UPF), 08002 Barcelona, Spain. [20]Department of Genetic Medicine and Development, University of Geneva Medical School, 1211 Geneva, Switzerland. [21]Institute for Genetics and Genomics in Geneva (iG3), University of Geneva, 1211 Geneva, Switzerland. [22]Swiss Institute of Bioinformatics, 1211 Geneva, Switzerland. [23]Department of Genetics, Perelman School of Medicine, University of Pennsylvania, Philadelphia, PA 19104, USA. [24]New York Genome Center, New York, NY 10013, USA. [25]Department of Systems Biology, Columbia University Medical Center, New York, NY 10032, USA. [26]Department of Public Health Sciences, The University of Chicago, Chicago, IL 60637, USA. [27]McDonnell Genome Institute, Washington University School of Medicine, St. Louis, MO 63108, USA. [28]Department of Genetics, Washington University School of Medicine, St. Louis, MO 63108, USA. [29]Department of Pathology & Immunology, Washington University School of Medicine, St. Louis, MO 63108, USA. [30]Department of Computer Science, Center for Statistics and Machine Learning, Princeton University, Princeton, NJ 08540, USA. [31]Department of Computer Science, University of California, Los Angeles, CA 90095, USA. [32]Department of Human Genetics, University of California, Los Angeles, CA 90095, USA. [33]Instituto de Investigação e Inovação em Saúde (i3S), Universidade do Porto, 4200-135 Porto, Portugal. [34]Institute of Molecular Pathology and Immunology (IPATIMUP), University of Porto, 4200-625 Porto, Portugal. [35]Division of Genetic Medicine, Department of Medicine, Vanderbilt University Medical Center, Nashville, Tennessee 37232, USA. [36]Department of Clinical Epidemiology, Biostatistics and Bioinformatics, Academic Medical Center, University of Amsterdam, 1105 AZ Amsterdam, The Netherlands. [37]Department of Psychiatry, Academic Medical Center, University of Amsterdam, 1105 AZ Amsterdam, The Netherlands. [38]Lewis Sigler Institute, Princeton University, Princeton, NJ 08540, USA.

[39]Department of Operations Research and Financial Engineering, Princeton University, Princeton, NJ 08540, USA. [40]Biomedical Informatics Program, Stanford University, Stanford, CA 94305, USA. [41]Institut Hospital del Mar d'Investigacions Mèdiques (IMIM), 08003 Barcelona, Spain. [42]Department of Medicine, Washington University School of Medicine, St. Louis, MO 63108, USA. [43]Department of Convergence Medicine, University of Ulsan College of Medicine, Asan Medical Center, Seoul 138-736, South Korea. [44]Department of Biomedical Engineering, Johns Hopkins University, Baltimore, MD 21218, USA. [45]Department of Biostatistics, Mailman School of Public Health, Columbia University, New York, NY 10032, USA. [46]Department of Biology, Stanford University, Stanford, CA 94305, USA. [47]Wellcome Trust Centre for Human Genetics, Nuffield Department of Medicine, University of Oxford, Oxford OX3 7BN, UK. [48]Oxford Centre for Diabetes, Endocrinology and Metabolism, University of Oxford, Churchill Hospital, Oxford OX3 7LE, UK. [49]Oxford NIHR Biomedical Research Centre, Churchill Hospital, Oxford OX3 7LJ, UK. [50]Computational Biology & Bioinformatics Graduate Program, Duke University, Durham, NC 27708, USA. [51]Human Genetics Department, McGill University, Montreal, QC H3A 0G1, Canada. [52]Departament d'Estadística i Investigació Operativa, Universitat Politècnica de Catalunya, 08034 Barcelona, Spain. [53]Department of Statistics and Operations Research, University of North Carolina, Chapel Hill, NC 27599, USA. [54]Department of Biostatistics, University of North Carolina, Chapel Hill, NC 27599, USA. [55]Section of Genetic Medicine, Department of Medicine, The University of Chicago, Chicago, IL 60637, USA. [56]Institute for Genomics and Systems Biology, The University of Chicago, Chicago, IL 60637, USA. [57]Department of Biostatistics, The University of Texas MD Anderson Cancer Center, Houston, TX 77030, USA. [58]Computational Sciences, Pfizer Inc, Cambridge, MA 02139, USA. [59]Universitat de Barcelona, 08028 Barcelona, Spain. [60]Department of Biomedical Data Science, Stanford University, Stanford, CA 94305, USA. [61]Department of Statistics, Stanford University, Stanford, CA 94305, USA. [62]Institute of Biophysics Carlos Chagas Filho (IBCCF), Federal University of Rio de Janeiro (UFRJ), 21941902 Rio de Janeiro, Brazil. [63]Department of Psychiatry, University of Utah, Salt Lake City, UT 84108, USA. [64]Department of Statistics, The University of Chicago, Chicago, IL 60637, USA. [65]Department of Human Genetics, The University of Chicago, Chicago, IL 60637, USA. [66]Center for Data Intensive Science, The University of Chicago, Chicago, IL 60637, USA. [67]Department of Psychiatry and Biobehavioral Sciences, University of California, Los Angeles, CA 90095, USA. [68]Department of Biostatistics, University of Michigan, Ann Arbor, MI 48109, USA. [69]Bioinformatics Research Center and Departments of Statistics and Biological Sciences, North Carolina State University, Raleigh, NC 27695, USA. [70]National Institute for Biotechnology in the Negev, Beer-Sheva 84105, Israel. [71]European Molecular Biology Laboratory, 69117 Heidelberg, Germany

## Enhancing GTEx (eGTEx) groups

Joshua M. Akey[38,72], Daniel Bates[73], Joanne Chan[14], Lin S. Chen[26], Melina Claussnitzer[10,74,75], Kathryn Demanelis[26], Morgan Diegel[73], Jennifer A. Doherty[76], Andrew P. Feinberg[44,77,78,79], Marian S. Fernando[55,56], Jessica Halow[73], Kasper D. Hansen[77,80,81], Eric Haugen[73], Peter F. Hickey[81], Lei Hou[10,82], Farzana Jasmine[26], Ruiqi Jian[14], Lihua Jiang[14], Audra Johnson[73], Rajinder Kaul[73], Manolis Kellis[10,82], Muhammad G. Kibriya[26], Kristen Lee[73], Jin Billy Li[14], Qin Li[14], Xiao Li[14], Jessica Lin[14,83], Shin Lin[14,84], Sandra Linder[14,15], Caroline Linke[55,56], Yaping Liu[10,82], Matthew T. Maurano[85], Benoit Molinie[10], Stephen B. Montgomery[14,15], Jemma Nelson[73], Fidencio J. Neri[73], Meritxell Oliva[55,56], Yongjin Park[10,82], Brandon L. Pierce[26], Nicola J. Rinaldi[10,82], Lindsay F. Rizzardi[77], Richard Sandstrom[73], Andrew Skol[55,56,66], Kevin S. Smith[14,15], Michael P. Snyder[14], John Stamatoyannopoulos[73,83,86], Barbara E. Stranger[55,56,66], Hua Tang[14], Emily K. Tsang[15,40], Li Wang[10], Meng Wang[14], Nicholas Van Wittenberghe[10], Fan Wu[55,56] & Rui Zhang[14]

[72]Department of Ecology and Evolutionary Biology, Princeton University, Princeton, NJ 08540, USA. [73]Altius Institute for Biomedical Sciences, Seattle, Washington 98121, USA. [74]Beth Israel Deaconess Medical Center, Harvard Medical School, Boston, MA 02215, USA. [75]University of Hohenheim, 70599 Stuttgart, Germany. [76]Huntsman Cancer Institute, Department of Population Health Sciences, University of Utah, Salt Lake City, UT 84112, USA. [77]Center for Epigenetics, Johns Hopkins University School of Medicine, Baltimore, MD 21205, USA. [78]Department of Medicine, Johns Hopkins University School of Medicine, Baltimore, MD 21205, USA. [79]Department of Mental Health, Johns Hopkins University School of Public Health, Baltimore, MD 21205, USA. [80]McKusick-Nathans Institute of Genetic Medicine, Johns Hopkins School of Medicine, Baltimore, MD 21205, USA. [81]Department of Biostatistics, Johns Hopkins University, Baltimore, MD 21205, USA. [82]Computer Science and Artificial Intelligence Laboratory, Massachusetts Institute of Technology, Cambridge, MA 02139, USA. [83]Department of Medicine, University of Washington, Seattle, Washington 98195, USA. [84]Division of Cardiology, University of Washington, Seattle, Washington 98195, USA. [85]Institute for Systems Genetics, New York University Langone Medical Center, New York, NY 10016, USA. [86]Department of Genome Sciences, University of Washington, Seattle, WA 98195, USA

## NIH Common Fund

Concepcion R. Nierras[87]

[87]Office of Strategic Coordination, Division of Program Coordination, Planning and Strategic Initiatives, Office of the Director, NIH, Rockville, MD 20852, USA

## NIH/NCI

Philip A. Branton[88], Latarsha J. Carithers[88,89], Ping Guan[88], Helen M. Moore[88], Abhi Rao[88] & Jimmie B. Vaught[88]

[88]Biorepositories and Biospecimen Research Branch, Division of Cancer Treatment and Diagnosis, National Cancer Institute, Bethesda, MD 20892, USA. [89]National Institute of Dental and Craniofacial Research, Bethesda, MD 20892, USA

## NIH/NHGrI

Sarah E. Gould[90], Nicole C. Lockart[90], Casey Martin[90], Jeffery P. Struewing[90] & Simona Volpi[90]

[90]Division of Genomic Medicine, National Human Genome Research Institute, Rockville, MD 20852, USA

## NIH/NIMH

Anjene M. Addington[91] & Susan E. Koester[91]

[91]Division of Neuroscience and Basic Behavioral Science, National Institute of Mental Health, NIH, Bethesda, MD 20892, USA

## NIH/NIDA

A. Roger Little[92]

[92]Division of Neuroscience and Behavior, National Institute on Drug Abuse, NIH, Bethesda, MD 20892, USA

## Biospecimen Collection Source Site—NDrI

Lori E. Brigham[93], Richard Hasz[94], Marcus Hunter[95], Christopher Johns[96], Mark Johnson[97], Gene Kopen[98], William F. Leinweber[98], John T. Lonsdale[98], Alisa McDonald[98], Bernadette Mestichelli[98], Kevin Myer[95], Brian Roe[95], Michael Salvatore[98], Saboor Shad[98], Jeffrey A. Thomas[98], Gary Walters[97], Michael Washington[97] & Joseph Wheeler[96]

[93]Washington Regional Transplant Community, Falls Church, VA 22003, USA. [94]Gift of Life Donor Program, Philadelphia, PA 19103, USA. [95]LifeGift, Houston, TX 77055, USA. [96]Center for Organ Recovery and Education, Pittsburgh, PA 15238, USA. [97]LifeNet Health, Virginia Beach, VA 23453, USA. [98]National Disease Research Interchange, Philadelphia, PA 19103, USA

## Biospecimen Collection Source Site—rPCI

Jason Bridge[99], Barbara A. Foster[100], Bryan M. Gillard[100], Ellen Karasik[100], Rachna Kumar[100], Mark Miklos[99] & Michael T. Moser[100]

[99]Unyts, Buffalo, NY 14203, USA. [100]Pharmacology and Therapeutics, Roswell Park Cancer Institute, Buffalo, NY 14263, USA

## Biospecimen Core resource—VArI

Scott D. Jewell[101], Robert G. Montroy[101], Daniel C. Rohrer[101] & Dana R. Valley[101]

[101]Van Andel Research Institute, Grand Rapids, MI 49503, USA

## Brain Bank repository—University of Miami Brain Endowment Bank

David A. Davis[102] & Deborah C. Mash[102]

[102]Brain Endowment Bank, Miller School of Medicine, University of Miami, Miami, FL 33136, USA

## Leidos Biomedical—Project Management

Anita H. Undale[103], Anna M. Smith[104], David E. Tabor[104], Nancy V. Roche[104], Jeffrey A. McLean[104], Negin Vatanian[104], Karna L. Robinson[104], Leslie Sobin[104], Mary E. Barcus[105], Kimberly M. Valentino[104], Liqun Qi[104], Steven Hunter[104], Pushpa Hariharan[104], Shilpi Singh[104], Ki Sung Um[104], Takunda Matose[104] & Maria M. Tomaszewski[104]

[103]National Institute of Allergy and Infectious Diseases, NIH, Rockville, MD 20852, USA. [104]Biospecimen Research Group, Clinical Research Directorate, Leidos Biomedical Research, Inc., Rockville, MD 20852, USA. [105]Leidos Biomedical Research, Inc, Frederick, MD 21701, USA

## ELSI Study

Laura K. Barker[106], Maghboeba Mosavel[107], Laura A. Siminoff[106] & Heather M. Traino[106]

[106]Temple University, Philadelphia, PA 19122, USA. [107]Department of Health Behavior and Policy, School of Medicine, Virginia Commonwealth University, Richmond, VA 23298, USA

## Genome Browser Data Integration & Visualization—EBI

Paul Flicek[108], Thomas Juettemann[108], Magali Ruffier[108], Dan Sheppard[108], Kieron Taylor[108], Stephen J. Trevanion[108] & Daniel R. Zerbino[108]

[108]European Molecular Biology Laboratory, European Bioinformatics Institute, Hinxton CB10 1SD, UK

## Genome Browser Data Integration & Visualization—UCSC Genomics Institute, University of California Santa Cruz

Brian Craft[109], Mary Goldman[109], Maximilian Haeussler[109], W. James Kent[109], Christopher M. Lee[109], Benedict Paten[109], Kate R. Rosenbloom[109], John Vivian[109] & Jingchun Zhu[109]

[109]UCSC Genomics Institute, University of California Santa Cruz, Santa Cruz, CA 95064, USA

