## [Peer Review File · Nature Communications]

Reviewers' comments:

Reviewer #1 (Remarks to the Author):

The authors have performed a thorough and diligent revision and the manuscript is much improved. I appreciate the detailed discussion and am generally convinced by their response to reviewers. The paper makes a useful contribution in presenting relationships between the Predixcan/TWAS/SMR statistics, contains compelling results in real traits (particularly the novel observation about enrichment of ClinVar genes), and presents a useful resource across many tissues and GWAS studies. I have a small number of remaining comments:

1. [previous comment 6] Thank you for clarifying the goal and details of the simulations in such detail, these results make much more sense to me now. I still find simulating ONLY under the null hypothesis to be somewhat incomplete; this shows that the S-PrediXcan statistic is not strongly susceptible to false-positives due to LD differences - which is important! But it still does not demonstrate that the statistic is valid under the alternative hypothesis (or the expected loss of power). I think it would be useful to demonstrate this using some standard simulations from the expected alternative (i.e. one or two SNPs effect expression, and expression increases disease risk).

A potentially bigger issue is that it actually looks like S-PrediXcan is yielding some false-positives in Figure 2B: the middle-left panel has an estimate of S-PrediXcan = -7 while PrediXcan = -2; and the middle-right panel has S-PrediXcan >4 for PrediXcan < 2.5 . Please comment on these, though I'll underscore that - due to differences in genetic architecture across traits - this is probably best quantified with simulations under the alternative.

2. In comparisons with COLOC (pg.5 and elsewhere), it's worth mentioning that COLOC also operates under a specific model assumption of at most one shared causal variant in the locus. The increased colocalization of Elastic Net relative to BSLMM may be therefore be reflective of EN identifying genes that are more consistent with the COLOC model, but not more biologically colocalized in truth. In general, the COLOC assumption is fairly strict given that there is substantial evidence of allelic heterogeneity in expression, so it's important to draw a distinction between statistical colocalization by COLOC and experimental colocalization that's out of scope of this work.

For the same reason, the proportion of colocalized loci is a somewhat misleading statistic. If S-PrediXcan finds 10 significant genes, and TWAS finds those 10 genes + 5 more genes that have evidence of heterogeneity or weaker signals (and therefore low COLOC statistics), then S-PrediXcan will come out on top. What does the raw # of colocalized genes across the two methods look like? I seem to recall the conclusion from the previous version of the manuscript to be that there is no clear winner between S-PrediXcan and TWAS, but I'm not seeing that in this version.

Finally, please add an additional plot to Figure 4 comparing the proportion of colocalization for SMR and S-PrediXcan.

3. I somewhat disagree with the statement "the relationship between the individual level and summary level based versions of TWAS was only reported in applications and simulations". Looking through the TWAS paper, the S-TWAS approach is explicitly cast as a summary-based genetic

imputation with Gaussian assumptions (which itself is previously proposed in Pasaniuc et al. 2014 Bioinformatics; and in a more basic form in Wen & Stephens Ann Appl Stat 2010). The S-TWAS formulation is that T_g is a latent genetic variant whose individual-level association to Y is imputed under a Gaussian model of SNPs. In contrast, the S-Predixcan formulation explicitly defines T_g as a linear combination of SNP weights and derives the relationship between the weights and the phenotype using summary statistics, while noting assumptions about statistical uncertainty. These are two different perspectives on the underlying quantity and the S-Predixcan generalization is useful. But while it is accurate to conclude, as previous versions of the manuscript did, that this is a generalized derivation of the statistic, I do not believe it is accurate to conclude that for TWAS (and previous work) this relationship was only reported in applications and simulations.

Minor comments:

* This is just a recommendation, but consider adding to the section starting on line 266 a reference to the recent perspective of Boyle et al. 2017 Cell on omnigenic disease architecture.

* line 167 should be "effect ON the gene"

* The Discussion is fairly terse and I'd recommend rewriting to improve the flow.

Reviewer #3 (Remarks to the Author):

This paper describes an extension to the PrediXcan framework that allows it to operate on summary statistics instead of full genotype data. Half of the paper is devoted to comparisons with TWAS and SMR, and the other half describes results for application to a large collection of human phenotypes. I have no further comments for the second half of the paper beyond those previously given. I have the following additional comments for the first half of the paper:

In the comparison with TWAS the authors state that, "We also point out the likely higher sensitivity to LD-contaminations when using prediction models with a large polygenic component such as BSLMM as implemented by TWAS." This claim is based on the application of Coloc to TWAS and PrediXcan results from real data sets. While it is semi-convincing the difficulty is that the underlying truth is not known. The authors should use simulation with single and multiple causal variants per gene and then see the extent to which the two methods are misspecified.

Variable were not defined before their use in multiple places making it difficult to read.

Line 121 R_l and R_g are not defined

Line 125-126 subscripts in variables are not defined.

The comparison to SMR assumes an intimate knowledge of the SMR paper in order to read that section.

Line 159-160 This is a difference in assumption but is not clear that you are correct. What if there was a distal effect of that SNP and no local effect? Perhaps say that your method is more powerful when your assumptions are correct and then state why your assumptions are more likely.

Line 184-185 Language like "state of the art methods" should be saved for the discussion.

Line 186 "mostly with elastic net model" what do you mean by "mostly"?

Reviewer #1 (Remarks to the Author):

The authors have performed a thorough and diligent revision and the manuscript is much improved. I appreciate the detailed discussion and am generally convinced by their response to reviewers. The paper makes a useful contribution in presenting relationships between the Predixcan/TWAS/SMR statistics, contains compelling results in real traits (particularly the novel observation about enrichment of ClinVar genes), and presents a useful resource across many tissues and GWAS studies.

We thank the reviewer for the time and effort to thoroughly review the multiple versions of our manuscript. We gratefully acknowledge that this process has substantially improved our manuscript.

We would like to mention that at the request of the authors of SMR, we have changed the null hypothesis used in the simulation of SMR. For the instrument to be valid, they consider that the null hypothesis should be that the eQTL Z_{score}^2 arises from a noncentral χ^2 , and only include eQTLs with p-values smaller than $10e-8$. We followed their request and still the statistic is deflated, with the mean centered around 0.93. We made changes to the manuscript reflecting this. These changes do not alter the message of our paper.

I have a small number of remaining comments:

1. [previous comment 6] Thank you for clarifying the goal and details of the simulations in such detail, these results make much more sense to me now. I still find simulating ONLY under the null hypothesis to be somewhat incomplete; this shows that the S-PrediXcan statistic is not strongly susceptible to false-positives due to LD differences - which is important! But it still does not demonstrate that the statistic is valid under the alternative hypothesis (or the expected loss of power). I think it would be useful to demonstrate this using some standard simulations from the expected alternative (i.e. one or two SNPs effect expression, and expression increases disease risk).

Action: We performed several simulations under the alternative hypothesis and generated **Supplementary Figure 11**, where we show a representative comparison of PrediXcan and S-PrediXcan under the alternative hypothesis.

A potentially bigger issue is that it actually looks like S-PrediXcan is yielding some false-positives in Figure 2B: the middle-left panel has an estimate of S-PrediXcan = -7 while PrediXcan = -2; and the middle-right panel has S-PrediXcan >4 for PrediXcan < 2.5. Please comment on these, though I'll underscore that - due to differences in genetic architecture across traits - this is probably best quantified with simulations under the alternative.

Response:

We agree with the reviewer that we should discuss this point and warn users of potential false positives. Although in general we find good correlation between summary and individual level based PrediXcan, there are a handful of mismatches as pointed out by the reviewer (African and East Asian populations, in that case). This can only be solved using the right correlation information and underscores the need to use matched population LD.

Action: We have added to the discussion the following paragraph to the Results (page 3, line 81) and the Discussion (page 14, line 370):

"Despite the generally good concordance between the summary and individual level methods, there were a handful of false positive results with S-PrediXcan much more significant than PrediXcan. This underscores the need to use closely matched LD information whenever possible."

2. In comparisons with COLOC (pg.5 and elsewhere), it's worth mentioning that COLOC also operates under a specific model assumption of at most one shared causal variant in the locus. The increased colocalization of Elastic Net relative to BSLMM may be therefore be reflective of EN identifying genes that are more consistent with the COLOC model, but not more biologically colocalized in truth. In general, the COLOC assumption is fairly strict given that there is substantial evidence of allelic heterogeneity in expression, so it's important to draw a distinction between statistical colocalization by COLOC and experimental colocalization that's out of scope of this work.

Response: We agree that the assumption of unique causal variant could be unfairly undermining the true colocalization of signals detected by BSLMM models. However, based on our findings that gene expression traits are driven mostly by sparse terms rather the infinitesimal (Wheeler et al, 2016), we believe that the polygenic component of BSLMM increases the vulnerability of its predictions to LD contamination without adding prediction accuracy. Also notice that the elastic net model does take into account allelic heterogeneity.

Action: We have added the following sentence in the Discussion (page 14, line 361):

"Improved colocalization methods without single causal variant assumption may be needed to strengthen this argument. But sparse genetic architecture of gene expression traits supports the benefit of elastic net over BSLMM predictors."

For the same reason, the proportion of colocalized loci is a somewhat misleading statistic. If S-PrediXcan finds 10 significant genes, and TWAS finds those 10 genes + 5 more genes that have evidence of heterogeneity or weaker signals (and therefore low COLOC statistics), then S-PrediXcan will come out on top. What does the raw # of colocalized genes across the two methods look like? I seem to recall the conclusion from the previous version of the manuscript to be that there is no clear winner between S-PrediXcan and TWAS, but I'm not seeing that in this version.

Response: PrediXcan yields a larger number of significant associations than was reported for TWAS in Mancuso et al probably due to the smaller number of genes tested in TWAS relative to PrediXcan. Mancuso et al use a criterion based on GCTA heritability, which we have shown to underestimate h^2 (Wheeler et al 2016 PLoS Genetics).

Action: We added supplementary figure 13 comparing the number of associations. The caption addresses the difference as follows: "Notice that Mancuso et al filtered out genes with low GCTA heritability, which we have shown to underestimate h^2 (Wheeler et al 2016 PLoS Genetics). This results in much smaller number of genes tested with TWAS than with PrediXcan. This in turn explains the smaller number of significant genes in TWAS despite the fact that when genes are tested the significance between the two methods are similar as seen in Figure 4B."

Finally, please add an additional plot to Figure 4 comparing the proportion of colocalization for SMR and S-PrediXcan.

Response: Please keep in mind that SMR includes HEIDI test as a second step.

Action: We added Supplementary Figure 12 displaying the requested comparison.

3. I somewhat disagree with the statement "the relationship between the individual level and summary level based versions of TWAS was only reported in applications and simulations". Looking through the TWAS paper, the S-TWAS approach is explicitly cast as a summary-based genetic imputation with Gaussian assumptions (which itself is previously proposed in Pasaniuc et al. 2014 Bioinformatics; and in a more basic form in Wen & Stephens Ann Appl Stat 2010). The S-TWAS formulation is that T_g is a latent genetic variant whose individual-level association to Y is imputed under a Gaussian model of SNPs.

In contrast, the S-Predixcan formulation explicitly defines T_g as a linear combination of SNP weights and derives the relationship between the weights and the phenotype using summary statistics, while noting assumptions about statistical uncertainty. These are two different perspectives on the underlying quantity and the S-Predixcan generalization is useful. But while it is accurate to conclude, as previous versions of the manuscript did, that this is a generalized derivation of the statistic, I do not believe it is accurate to conclude that for TWAS (and previous work) this relationship was only reported in applications and simulations.

Answer: We agree with the reviewer's description of the imputation approach which uses Gaussian underlying model (BLUP). However, when the authors extend their approach to general weights (such as BSLMM), it is no longer interpretable as imputation and the relationship with the individual level approach is not obvious. Our derivation makes explicit how the two approaches (general weights TWAS statistic and individual level TWAS) are related.

Action: We modified the wording in Discussion (page 13, line 356) as follows:

"When Gaussian imputation is used, the relationship between individual level and summary versions of TWAS is clear. This is not the case when extended to general weights. Our mathematical derivation shows the analytic difference between them explicitly."

Minor comments:

* This is just a recommendation, but consider adding to the section starting on line 266 a reference to the recent perspective of Boyle et al. 2017 Cell on omnigenic disease architecture.

Action: We have added the following sentence to page 12 (line 337):

"The high level of concordance is supportive of a omnigenic trait architecture \cite{boyle2017}"

* line 167 should be "effect ON the gene"

Action: Fixed wording

* The Discussion is fairly terse and I'd recommend rewriting to improve the flow.

Reponse: Given the word limits and the need to address reviewers' comments and recommendations, we find it hard to change the wording of the discussion. If changing this is not completely necessary, we would prefer to avoid further editing the discussion.

Reviewer #3 (Remarks to the Author):

We would like to mention to the reviewer that at the request of the authors of SMR, we have changed the null hypothesis used in the simulation of SMR. For the instrument to be valid, they consider that the null hypothesis should be that the eQTL Z_{score}^2 arises from a noncentral χ^2 , and only include eQTLs with p-values smaller than $10e-8$. We followed their request and still the statistic is deflated, with the mean centered around 0.93. We made changes to the manuscript reflecting this. These changes do not alter the message of our paper.

This paper describes an extension to the PrediXcan framework that allows it to operate on summary statistics instead of full genotype data. Half of the paper is devoted to comparisons with TWAS and SMR, and the other half describes results for application to a large collection of human phenotypes. I have no further comments for the second half of the paper beyond those previously given. I have the following additional comments for the first half of the paper:

In the comparison with TWAS the authors state that, "We also point out the likely higher sensitivity to LD-contaminations when using prediction models with a large polygenic component such as BSLMM as implemented by TWAS." This claim is based on the application of Coloc to TWAS and PrediXcan results from real data sets. While it is semi-convincing the difficulty is that the underlying truth is not known. The authors should use simulation with single and multiple causal variants per gene and then see the extent to which the two methods are misspecified.

Response: We agree with the reviewer that the underlying truth is not known in real data.

Action: We have added the following sentence in the Discussion (page 14, line 361):

"Improved colocalization methods without single causal variant assumption may be needed to strengthen this argument. But sparse genetic architecture of gene expression traits supports the benefit of elastic net over BSLMM predictors."

Variable were not defined before their use in multiple places making it difficult to read.

Line 121 R_I and R_g are not defined

Line 125-126 subscripts in variables are not defined.

Action: We added a short explanation in those lines.

The comparison to SMR assumes an intimate knowledge of the SMR paper in order to read that section.

Line 159-160 This is a difference in assumption but is not clear that you are correct. What if there was a distal effect of that SNP and no local effect? Perhaps say that your method is more powerful when your assumptions are correct and then state why your assumptions are more likely.

Response: We should clarify that both SMR and PrediXcan (and TWAS) use cis-associations only. So there is no difference in underlying assumptions in this respect.

Action: We have added the following (page 6, line 160):

Keep in mind that currently both SMR and PrediXcan only use cis associations."

Line 184-185 Language like "state of the art methods" should be saved for the discussion.

Action: We have eliminated "state of the art" from the sentence

Line 186 "mostly with elastic net model" what do you mean by "mostly"?

Action: eliminated "mostly" from the sentence

REVIEWERS' COMMENTS:

Reviewer #1 (Remarks to the Author):

Thank you for responding to my comments and performing additional simulations, my concerns are largely addressed except for one presentation choice (below). I appreciate your careful revisions throughout this process.

Supplementary Figure 12 should be a part of main Figure 4 as it is a relevant comparison between the three methods and yields an informative result: SMR appears to be much more variable than PrediXcan, identifying both a greater % co-localized and a greater % non-colocalized genes. Currently the analysis is presented without description, but I believe this is an important additional result to discuss in the main text.

Supplementary Figures 1-5 are quite interesting but are not referenced anywhere in the text (the first reference is Supplementary Figure 11). I don't have a recommendation but I hope readers are made aware of these analyses.

Referee 1's additional feedback on referee 2's last concern on sensitivity to LD-contaminations when using prediction models:

The reviewer makes one major comment, recommending "The authors should use simulation with single and multiple causal variants per gene and then see the extent to which the two methods are misspecified." . The authors do not perform additional simulations, but only add a sentence to the Discussion outlining this issue. Strictly speaking the author changes are not responsive.

I personally don't find the new discussion point to be sufficiently convincing to justify not performing the simulations for several reasons: (1) there is growing evidence that genes often have multiple causal variants (for example GTEx Consortium 2017 Nature identifying 41.2% of genes with conditionally significant secondary effects), and more than a single eQTL would be enough to violate the COLOC model; (2) the authors claim that elastic net is better for sparse architectures than BSLMM but BSLMM is also a sparse predictor (that's what the "S" stands for) and is, in fact, the model used to quantify the sparse architecture of gene expression in ref.27 so it's counterintuitive that this model would be a poor predictor; (3) the eCAVIAR method of Hormozdiari et al. 2016 AJHG is a popular colocalization method that models multiple causal variants, so there are methods currently available to quantify this.

I think there are a few ways to address this concern: (1) do the simulations recommended by the reviewer with multiple causal eQTLs and show that elastic net is still a better predictor than BSLMM, and that COLOC provides an accurate assessment of colocalization even though it's model is violated; (2) remove the concerning statements in the Discussion (p.14) that TWAS/BSLMM is more susceptible to LD-contamination, remove the statement that elastic net performs better than BSLMM, remove the statement (p.6) that TWAS has fewer colocalized signals than PrediXcan because of contamination, and instead discuss the caveat that genes with multiple eQTLs are a model violation for COLOC and therefore no definitive conclusion can be drawn from the improved colocalization of PrediXcan signals.

REVIEWERS' COMMENTS:

Reviewer #1 (Remarks to the Author):

Thank you for responding to my comments and performing additional simulations, my concerns are largely addressed except for one presentation choice (below). I appreciate your careful revisions throughout this process.

Supplementary Figure 12 should be a part of main Figure 4 as it is a relevant comparison between the three methods and yields an informative result: SMR appears to be much more variable than PrediXcan, identifying both a greater % co-localized and a greater % non-colocalized genes. Currently the analysis is presented without description, but I believe this is an important additional result to discuss in the main text.

Action: Following the reviewer's advice, we moved Supp figure 12 into Figure 5. Figure 4 shows the comparison between TWAS and S-PrediXcan. And Figure 5 is the comparison with SMR. We added the following text to the main text:

"Fig. 5g shows a comparison between SMR's and S-PrediXcan's proportion of non-colocalization, while Fig. 5h compares proportion of colocalization, as estimated by COLOC. SMR shows a higher proportion of colocalized and independent signals. This is expected since SMR uses more stringent eQTL association criterion so that there are few significant genes in the undetermined region."

Supplementary Figures 1-5 are quite interesting but are not referenced anywhere in the text (the first reference is Supplementary Figure 11). I don't have a recommendation but I hope readers are made aware of these analyses.

Response: Supplementary Figure 5 is covered in part of the section **Agnostic scanning across GTEx tissues improves discovery**. We added an overall description of Supplementary Figures 1-4 to the main text.

Action: We added a brief overview of Supplementary Figures 1-4 in section **Gene expression variation is associated to diverse traits**.

"S-PrediXcan results tend to be more significant as the genetic component of gene expression increases (larger cross-validated prediction performance). Similarly, S-PrediXcan associations tend to be more significant when prediction is more reliable (p-values of association between predicted and observed expression levels are more significant, i.e. when prediction performance p-value is smaller). The trend is seen both when results are averaged across all tissues for a

given phenotype or across all phenotypes for a given tissue, as displayed in Supplementary Figures 1-4. This trend was also robust across different monotonic functions of the Z-scores.”

Referee 1’s additional feedback on referee 2’s last concern on sensitivity to LD-contaminations when using prediction models:

The reviewer makes one major comment, recommending “The authors should use simulation with single and multiple causal variants per gene and then see the extent to which the two methods are misspecified.” . The authors do not perform additional simulations, but only add a sentence to the Discussion outlining this issue. Strictly speaking the author changes are not responsive.

I personally don’t find the new discussion point to be sufficiently convincing to justify not performing the simulations for several reasons: (1) there is growing evidence that genes often have multiple causal variants (for example GTEx Consortium 2017 Nature identifying 41.2% of genes with conditionally significant secondary effects), and more than a single eQTL would be enough to violate the COLOC model; (2) the authors claim that elastic net is better for sparse architectures than BSLMM but BSLMM is also a sparse predictor (that’s what the “S” stands for) and is, in fact, the model used to quantify the sparse architecture of gene expression in ref.27 so it’s counterintuitive that this model would be a poor predictor; (3) the eCAVIAR method of Hormozdiari et al. 2016 AJHG is a popular colocalization method that models multiple causal variants, so there are methods currently available to quantify this.

I think there are a few ways to address this concern: (1) do the simulations recommended by the reviewer with multiple causal eQTLs and show that elastic net is still a better predictor than BSLMM, and that COLOC provides an accurate assessment of colocalization even though it’s model is violated; (2) remove the concerning statements in the Discussion (p.14) that TWAS/BSLMM is more susceptible to LD-contamination, remove the statement that elastic net performs better than BSLMM, remove the statement (p.6) that TWAS has fewer colocalized signals than PrediXcan because of contamination, and instead discuss the caveat that genes with multiple eQTLs are a model violation for COLOC and therefore no definitive conclusion can be drawn from the improved colocalization of PrediXcan signals.

Response:

We consider that proving robustness of COLOC is beyond the scope of this paper. Therefore, we have toned down the "controversial" statements as described below and added a discussion on COLOC’s limitations.

Action:

- Rewrote the sentence in the TWAS/S-PrediXcan comparison as: “We believe this is due to the polygenic component of BSLMM models, a wider set of SNPs increasing the chances of COLOC yielding a non-colocalized result”

- Changed:
 - “proportion of non-colocalized...” to “COLOC-estimated proportion of non-colocalized...”
 - “proportion of colocalized...” to “COLOC-estimated proportion of colocalized...”
- Moved the colocalization discussion to an earlier part in the discussion and explicitly added a caveat about COLOC assumptions: “COLOC has the limitation of assuming a single causal variant, and has reduced power in the presence of multiple causal variants.”
- We removed the statement about BSLMM and rewrote the appropriate paragraph in the discussion as:

“We find that BSLMM-based TWAS results have a larger proportion of non-colocalized genes as estimated by COLOC. This could be due to the single variant assumption in COLOC but we believe this is rather a consequence of the polygenic component of BSLMM predictors. Given the predominantly sparse architecture of gene expression traits ²⁷, we believe that adding a polygenic component unnecessarily increases the exposure to LD-contamination”